# *Plasmodium falciparum* egress disrupts endothelial junctions and activates JAK-STAT signaling in a microvascular 3D blood-brain barrier model

Livia Piatti [1,9], Alina Batzilla[1,2,9], Fumio Nakaki [1], Hannah Fleckenstein[1,3], François Korbmacher[1], Rory K. M. Long[1,2], Daniel Schraivogel[4], John A. Hawkins [4], Tais Romero-Uruñuela [1,8], Borja López-Gutiérrez [1], Silvia Sanz Sender [1], Yannick Schwab [3], Lars M. Steinmetz[4,5,6], James Sharpe[1,7] & Maria Bernabeu [1] ✉

Cerebral malaria is a severe neurovascular complication of *Plasmodium falciparum* infection, with high mortality rates even after treatment with effective antimalarials. Limitations in current experimental models have hindered our knowledge of the disease. We developed a 3D blood-brain barrier (BBB) model with enhanced barrier properties using primary brain endothelial cells, astrocytes, and pericytes. Exposure to parasite egress products increases microvascular permeability, likely due to transcriptional downregulation of junctional and vascular development genes in endothelial cells. In addition, it increases the expression of ferroptosis markers, antigen presentation and type I interferon genes and upregulates the JAK-STAT pathway across all BBB cell types. Incubation with cytoadherent schizont-stage *P. falciparum*-infected erythrocytes induces a similar, but highly localized transcriptional shift, along with inter-endothelial gaps at sites of parasite egress, leading to enhanced permeability. Treatment with the JAK-STAT inhibitor Ruxolitinib prevents the increase in permeability induced by *P. falciparum* egress products. These findings provide key insights into the parasite-mediated mechanisms driving brain microvascular pathogenesis in cerebral malaria and suggest potential avenues for adjunctive therapies.

*Plasmodium falciparum* infections account for the majority of the 600,000 annual malaria deaths worldwide[1], with cerebral malaria (CM) being one of the deadliest complications. Histological examination of post-mortem brain samples from CM patients has identified the sequestration of *P. falciparum*-infected red blood cells (iRBC) in the brain microvasculature as a key disease hallmark, often accompanied by vascular pathology and endothelial dysfunction[2–4]. Recent magnetic resonance imaging (MRI) studies suggest that fatal brain swelling in

[1]European Molecular Biology Laboratory (EMBL) Barcelona, Barcelona, Spain. [2]Collaboration for joint PhD degree between EMBL and Heidelberg University, Faculty of Biosciences, Heidelberg, Germany. [3]European Molecular Biology Laboratory (EMBL), Cell Biology and Biophysics Unit, Heidelberg, Germany. [4]European Molecular Biology Laboratory (EMBL), Genome Biology Unit, Heidelberg, Germany. [5]Department of Genetics, Stanford University School of Medicine, Stanford, CA, USA. [6]Stanford Genome Technology Center, Palo Alto, CA, USA. [7]Institució Catalana de Recerca i Estudis Avançats (ICREA), Barcelona, Spain. [8]Present address: ISGlobal, Hospital Clínic, Universitat de Barcelona, Barcelona, Spain. [9]These authors contributed equally: Livia Piatti, Alina Batzilla. ✉e-mail: maria.bernabeu@embl.es

pediatric CM patients likely results from blood-brain barrier (BBB) dysfunction, disrupting the selective transport of fluids and molecules from blood vessels to the brain parenchyma and leading to vasogenic edema[5,6]. Even with treatments that rapidly clear parasites from blood[7], CM still has a 15–20% mortality rate[8], and half of the survivors suffer from long-term neurological and behavioral sequelae[9,10]. A better mechanistic understanding of how *P. falciparum* disrupts the BBB is crucial for developing adjunctive host-targeted treatments that could prevent deaths and long-term disabilities.

Two key parasite-induced disruptive mechanisms have been proposed. The first involves parasite binding to endothelial receptors, endothelial protein C receptor (EPCR)[11] and intercellular adhesion molecule 1 (ICAM-1)[12–14], and blockade of EPCR homeostatic functions[15,16]. This pathogenic mechanism is exclusive to *P. falciparum* and does not occur in other malaria species, including rodent malaria models. The second mechanism suggests that parasites egressing from iRBC release endothelial-disruptive products, such as heme[17], hemozoin[18], parasite histones[19,20], *P. falciparum* histidine-rich protein 2 (PfHRP2)[21], or glycophosphatidyl inositol (GPI)[22]. Previous studies on these two mechanisms have been conducted using in vitro endothelial-only cultures with reduced barrier properties. Nevertheless, the BBB is a multicellular interface whose properties arise from the physical and cellular cross-talk between endothelial cells, pericytes and astrocytes. Furthermore, mechanical cues from blood flow and the extracellular matrix enhance its barrier function[23].

Here, we develop a bioengineered microvascular model that incorporates all these components, resulting in improved barrier properties. The use of this model reveals that parasite products released during egress are responsible for an increase in vascular permeability, likely as a consequence of downregulation of endothelial junctional and vascular development pathways. We further demonstrate that *P. falciparum* egress products elicit the activation of inflammatory, including JAK-STAT, and antigen presentation pathways in all the cells that compose the BBB. When experiments are performed with cytoadhesive *P. falciparum*-iRBC, the disruptive effects are locally confined to regions of high sequestration and parasite egress. Yet, they still lead to an increase in BBB permeability. Ruxolitinib, an inhibitor of the JAK-STAT pathway, prevents *P. falciparum*-induced microvascular leakage, highlighting a link between inflammatory and vascular disruptive pathways.

## Results

### Primary brain astrocytes and pericytes improve endothelial barrier function

To determine the barrier-disruptive pathways of *P. falciparum* in the brain microvasculature, we developed a bioengineered 3D-BBB model. The microvascular model was fabricated in a type I collagen hydrogel, pre-patterned by a combination of soft lithography and injection molding that generates a microfluidic 13 × 13 grid[24,25]. Commercial primary human astrocytes and brain vascular pericytes were seeded in the bulk collagen solution, and primary human brain microvascular endothelial cells were seeded under gravity-driven flow into the microfluidic network (Fig. 1a). The identity of the three cell types was confirmed by the expression of specific markers including platelet endothelial cell adhesion molecule 1 (PECAM-1), von Willebrand factor (vWF), vascular endothelial (VE)-cadherin and β-catenin for endothelial cells, platelet-derived growth factor receptor β (PDGFRβ) and nerve/glial antigen 2 (NG2) for pericytes and glial fibrillary acidic protein (GFAP), S100B and aquaporin 4 (AQP4) for astrocytes (Supplementary Fig. 1a). After two days in culture, endothelial cells line the perfusable microvessels and are surrounded by a second layer of astrocytes and pericytes that sparsely contact the endothelium, resembling the 3D organization and architecture of the in vivo BBB (Fig. 1b). Generally, pericytes and astrocytes ensheath the abluminal side of microvessels, and collagen-residing astrocytes occasionally extend their end-feet towards endothelial cells (Fig. 1c, d). Quantitative RT-PCR

measurements comparing two different astrocyte-to-pericyte ratios revealed that co-culture with astrocytes and pericytes at a 7:3 ratio increased the expression of endothelial BBB-specific markers over time, including tight junction markers (*OCLN, CLDN5*), BBB-transporters (*LRP1, SLC* family) and efflux pumps (*PGP, ABC* family), peaking after 7 days in culture (Supplementary Fig. 1b). Notably, the increased expression of BBB markers was associated with an improvement in microvascular barrier properties. The 3D-BBB model containing pericytes and astrocytes showed a significant decrease in permeability to 70 kDa FITC-dextran ($2.15 \times 10^{-6}$ cm/s – Interquartile range (IQR) = 0.30, 6.37), compared to an endothelial-only model ($8.31 \times 10^{-6}$ cm/s – IQR = 3.98, 12.40) (Fig. 1e and Supplementary Fig. 1c). Thus, the enhanced permeability properties of our bioengineered 3D-BBB model provide a unique opportunity to study the mechanisms of vascular barrier disruption induced by *P. falciparum*.

### *P. falciparum*-iRBC products released during egress downregulate endothelial junction expression and increase microvascular permeability

The egress of *P. falciparum*-iRBC has long been recognized as an endothelial disruptive event[18–22,26]. However, previous studies were done in models with a weaker vascular barrier and did not include astrocytes or pericytes[27–30]. We first explored how parasite egress products contribute to malaria pathogenesis within our newly engineered 3D-BBB model. To this end, we generated a solution containing *P. falciparum* products released upon the egress of $5 \times 10^7$ tightly synchronized iRBC/mL (hereafter referred to as iRBC-egress media)[31] (Supplementary Fig. 2a). We estimate that this concentration of products is equivalent to $5 \times 10^4$ parasites/μL, levels often found in CM patients[5], however, this value is an approximation and may exceed peripheral parasite densities encountered in clinical infections. 3D-BBB microvessels were incubated with iRBC-egress media for 24 h and subjected to a multimodal analysis. This included single-cell RNA sequencing (scRNA-seq) on dissociated microvessels, electron and confocal microscopy on fixed microvessels, as well as live permeability measurements (Fig. 2a). As a control, we perfused the supernatant of an uninfected erythrocyte (uRBC) control processed in the same way as the infected counterpart.

The 3D-BBB microvessels were disassembled and dissociated into a single-cell suspension, followed by sample-multiplexing using the MULTI-seq protocol[32] to pool infected and control conditions for scRNA-seq. The 6454 quality-controlled cells in the resulting scRNA-seq dataset were visualized using a uniform manifold approximation and projection (UMAP) algorithm, revealing 7 distinct clusters after unsupervised clustering (Fig. 2b). Cell type annotation showed the presence of two clusters with high expression of endothelial markers (cluster 1 and 2: *CDH5, VWF* and *PECAM1*), 4 clusters expressing pericyte markers (clusters 4 and 6: high expression of *PDGFRB*, and clusters 3 and 5: moderate *PDGFRB*) and a cluster expressing astrocytic markers (cluster 7: *S100B* and *GFAP*) (Fig. 2b, c and Supplementary Fig. 2b). A shift in the transcriptional profile of the three cell types was observed upon exposure to iRBC-egress media, with endothelial cells splitting into a separate cluster from the control population (Fig. 2d). Differential gene expression analysis on endothelial cells revealed that iRBC-egress media caused a significant decrease in the expression of multiple genes encoding proteins that are critical for BBB integrity, including the tight junction genes *CLDN5* (Claudin-5) and *TJP1* (ZO-1) and the adherens junction transcript *CDH5* (VE-cadherin) (Fig. 2e and Supplementary Data 1). Gene ontology term (GO-term) over-representation analysis indicated that most downregulated transcripts in endothelial cells are associated with endothelial cell junctions and adhesion, cytoskeleton organization, blood vessel development, DNA repair, chromatin organization and Wnt signaling (Fig. 2f and Supplementary Data 2). We employed the *CellChat* package[33] to analyze differential ligand-receptor interactions among the three cell types present in the model after exposure to iRBC-egress media. The

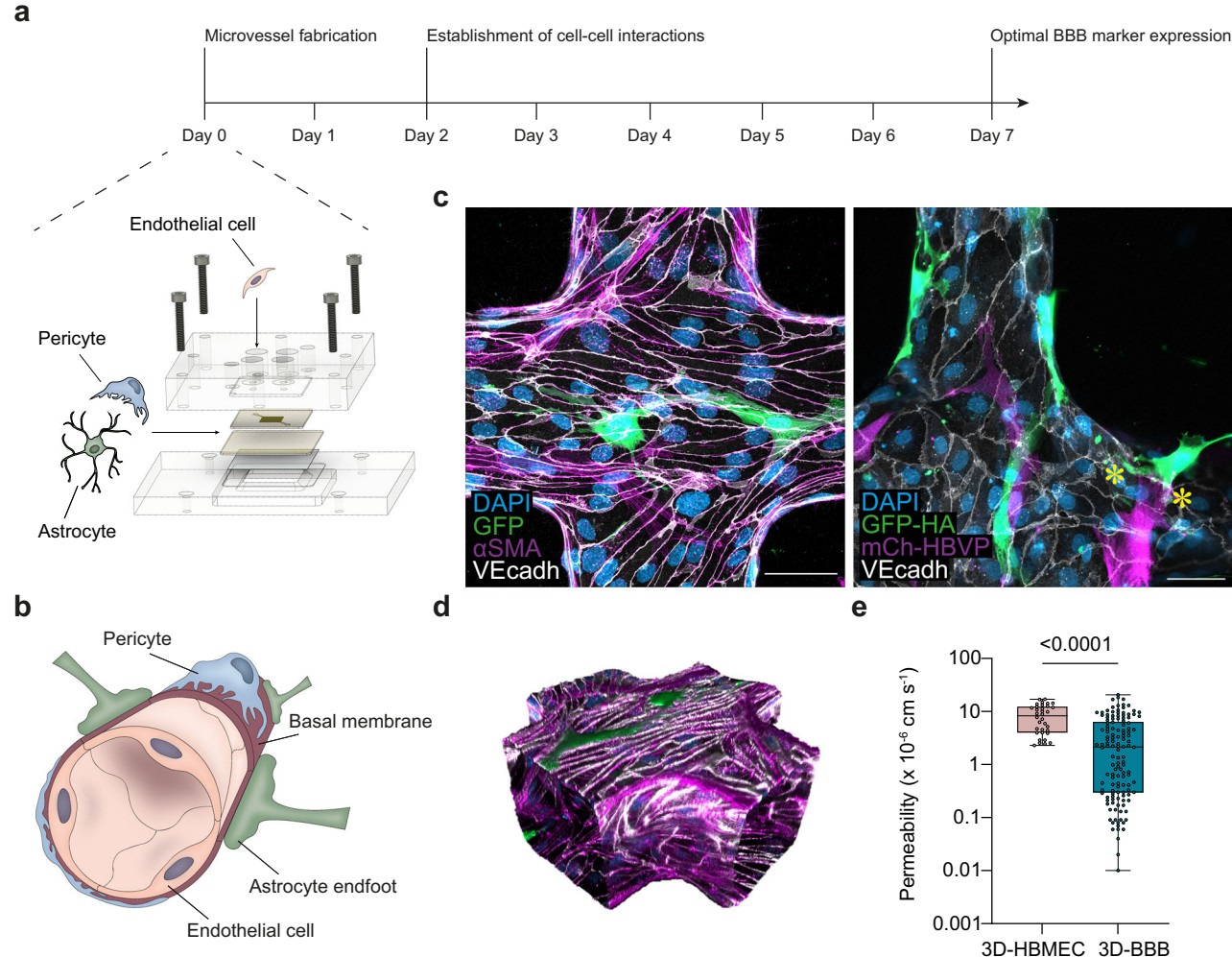

**Fig. 1 | Perfusable 3D-BBB microvascular model recapitulates the BBB structure, cell identities and presents improved barrier properties. a** Representation of the experimental timeline for 3D-BBB microvessel fabrication and culture, including a schematic sketch showing the materials for microvessel fabrication and the cells that compose it: human primary astrocytes, pericytes and brain microvascular endothelial cells. **b** Schematic depiction of the BBB architecture. **c** Maximum z-projection images of the 3D-BBB microvessel model and the spatial multicellular organization of endothelial cells stained with VE-cadherin (white), GFP-expressing astrocytes and pericytes labeled with αSMA (pink) (left image) or expressing mCherry (right image). Asterisks indicate astrocyte end feet contacting endothelial cells. Representative images from at least 3 independent experiments with similar results. **d** 3D reconstruction of a portion of the microfluidic network. **e** Apparent microvascular permeability to 70 kDa FITC-dextran. Each point represents an ROI from endothelial-only ($N = 3$) and 3D-BBB ($N = 13$) microvessels (Two-tailed Mann-Whitney U test). Box plots display the median (line within the box), IQR (box), and range (whiskers). Data are provided as a Source Data file.

analysis showed a decrease in vascular signaling interactions (*VWF*, *EDN1*), as well as in important pathways for endothelial-pericyte homeostasis, including signaling of angiopoietin-1 (*ANGPT1*), PDGF-BB (*PDGFB)*, and CSPG4 (*NG2*). In addition, iRBC-egress media caused a decrease in key endothelial-pericyte-astrocyte signaling molecules, including the Notch pathway (*DLL4, JAG1/2*) (Fig. 2g and Supplementary Fig. 3). Overall, exposure to iRBC-egress media led to the downregulation of transcripts associated with endothelial integrity and impaired major signaling pathways among all cell types present in the model.

To test whether these transcriptional changes resulted in an impairment of 3D microvascular integrity, we first quantified the number of inter-endothelial gaps, as well as the percentage of junctional length covered by electron-dense tight junctions, by transmission electron microscopy (TEM). Parasite egress products were visible by TEM and appeared as fuzzy aggregates containing parasite organelles and hemozoin, as well as parasitophorous and iRBC membranes with knobs (Fig. 3a). Although we observed a significantly higher percentage of ultrastructural gaps in 3D-BBB microvessels exposed to iRBC-egress media, we did not observe any significant differences in

the length of tight junctions in regions of cell-cell contact that remained intact (Fig. 3b). Changes in junction morphology were also observed by confocal microscopy. Specifically, VE-cadherin labeling revealed an altered adherens junction pattern compared to the control condition, with thin junctional staining and the formation of large inter-endothelial gaps (Fig. 3c). The transcriptional and morphological changes observed were accompanied by functional changes in the vascular barrier. Baseline microvascular permeability was measured by 70 kDa FITC-dextran perfusion on day 6 of 3D-BBB microvessel formation, followed by a second measurement in the same regions of interest 24 h after addition of iRBC-egress or control media (Fig. 3d). After 24 h, 3D-BBB microvessels treated with iRBC-egress media presented a significant 6.5-fold increase in microvascular permeability ratio (11.07 – IQR = 2.60, 32.56) compared to controls (1.68 – IQR = 0.87, 4.28) (Fig. 3e). Notably, no significant differences in microvascular permeability ratio were observed between the two HBMEC donors used in this study (Supplementary Fig. 2c), nor in 3D-BBB microvessels incubated with uRBC media control (Supplementary Fig. 2d). These results suggest that *P. falciparum*-iRBC products

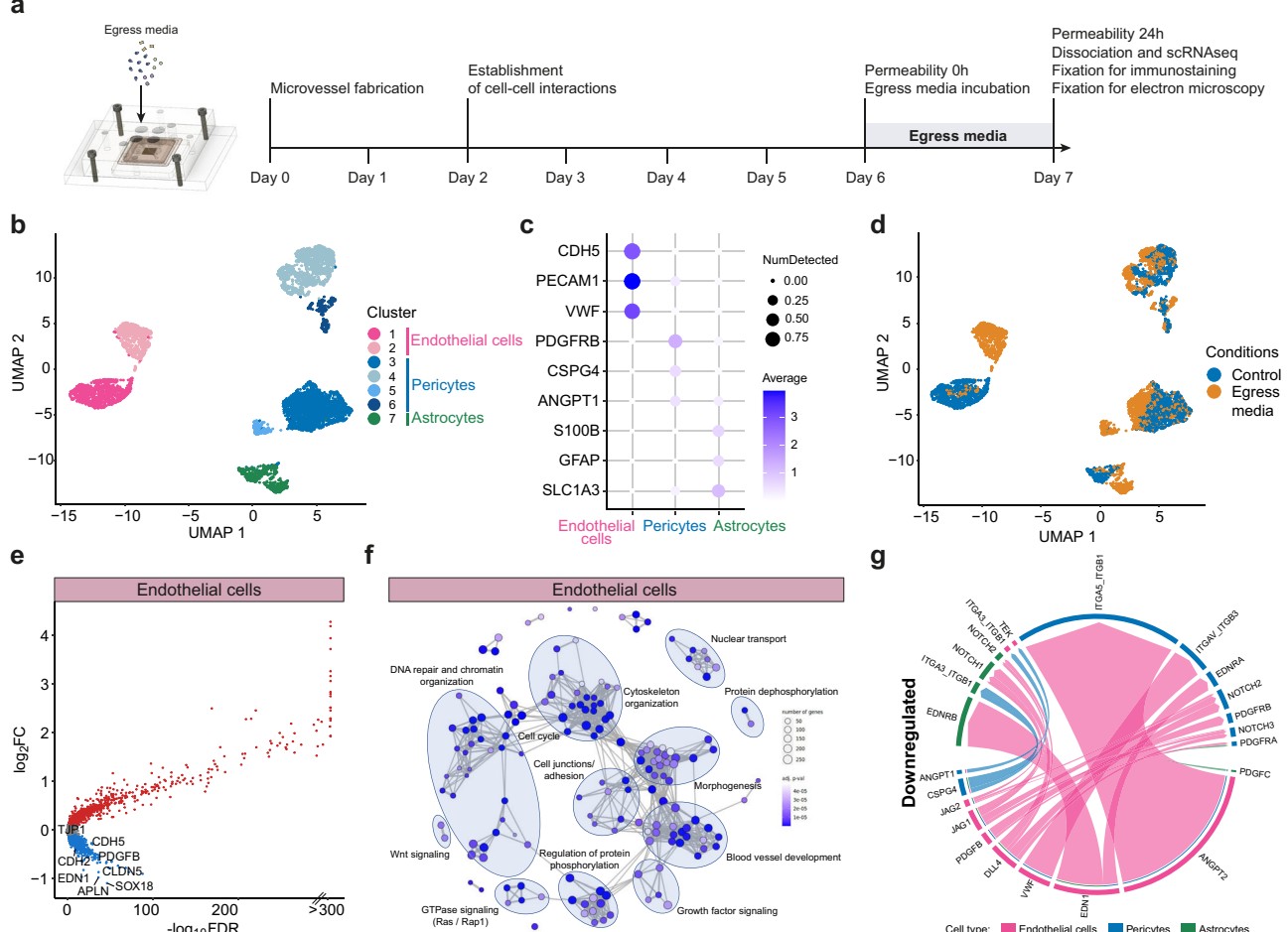

**Fig. 2 | *P. falciparum* iRBC-egress media induces transcriptional down-regulation of junctional markers and BBB signaling. a** Representation of the experimental timeline on 3D-BBB microvessels before and after 24-hour incubation with iRBC-egress media. **b**–**g** Single-cell transcriptomic analysis comparing 3D-BBB models exposed to iRBC-egress media and control uRBC media. **b** UMAP of sequenced cells colored by unsupervised *Leiden* clustering. **c** Dot plot of main BBB cell type markers. **d** UMAP of sequenced cells colored by experimental condition. **e** Volcano plot of differentially expressed genes in endothelial cells upon 24 h iRBC-egress media incubation, plotting the log2-transformed fold change (log2FC) against the statistical significance (-log10 of the false discovery rate (FDR)). Significantly up- or downregulated genes (FDR < 0.05, log2FC > 0.1 or log2FC < − 0.1, respectively) are marked in red or blue and selected downregulated genes are

labeled. **f** GO-term over-representation analysis on significantly downregulated genes (FDR < 0.05, log2FC < − 0.1) in endothelial cells. Each network node represents one of the most significant GO-terms (adjusted *p*-value < 0.0001), and edges connect GO-terms with more than 20% gene overlap. GO-term clusters were manually summarized with one label term. *P*-values were calculated using the hypergeometric distribution (one-sided Fisher's exact test). Multiple comparisons adjustment was performed using the Benjamini-Hochberg procedure. **g** Selection of downregulated ligand-receptor interactions important for BBB establishment, identified among the three BBB cell types after exposure to iRBC-egress media using the *CellChat* package. Arrows point from ligands on sender cells to receptors on receiver cells and are colored by the sender cell. Weights of links are proportional to the interaction strength.

decrease the expression of tight and adherens junction markers, changing the morphology of inter-endothelial junctions, together with a functional impairment of vascular barrier integrity.

### *P. falciparum* egress products induce a global activation of inflammatory, JAK-STAT, antigen presentation and ferroptosis-associated pathways in the 3D-BBB model

Studies in *P. berghei* rodent CM models have shown the ability of endothelial cells to cross-present parasite antigens present in merozoites[34,35]. Our analysis revealed that *P. falciparum* could induce a similar behavior in all the cells that compose our bioengineered BBB model. Specifically, exposure of 3D-BBB microvessels to iRBC-egress media caused the upregulation of multiple genes associated with inflammatory and antigen presentation pathways in endothelial cells (Fig. 4a). The same upregulated transcripts were identified in pericytes and astrocytes, suggesting that iRBC-egress media has the potential to cross the endothelial barrier. We found an upregulation

of transcripts involved in type I interferon (IFN) response and anti-viral pathways, including genes of the IFN-induced protein with tetratricopeptide repeats (IFIT) gene family (e.g., *IFIT1, IFIT2, IFIT3*), IFN-stimulated genes (ISGs) (e.g., *ISG15, ISG20*), and other IFN-inducible genes (e.g., *MX1, IFI6, IFI27, OAS1*), as well as ferroptosis genes (e.g., *HMOX1*) (Fig. 4a). Furthermore, transcripts upregulated by *P. falciparum* egress products include members of the JAK-STAT family of signal transducers (*STAT1, STAT2, JAK2*), a signaling pathway that induces the expression of ISGs, and some of their interacting proteins (*IRF1, IRF9*). GO-term over-representation analysis on the significantly upregulated transcripts confirmed a global increase in expression of transcripts associated with cytokine, viral, and type I IFN response, as well as antigen presentation, NFκB signaling, and protein catabolism in all cell types that compose the BBB (Fig. 4b and Supplementary Data 2). Cell type-specific processes that were significantly upregulated include apoptosis and autophagy in endothelial cells and astrocytes, ER stress and Golgi/vesicle transport in

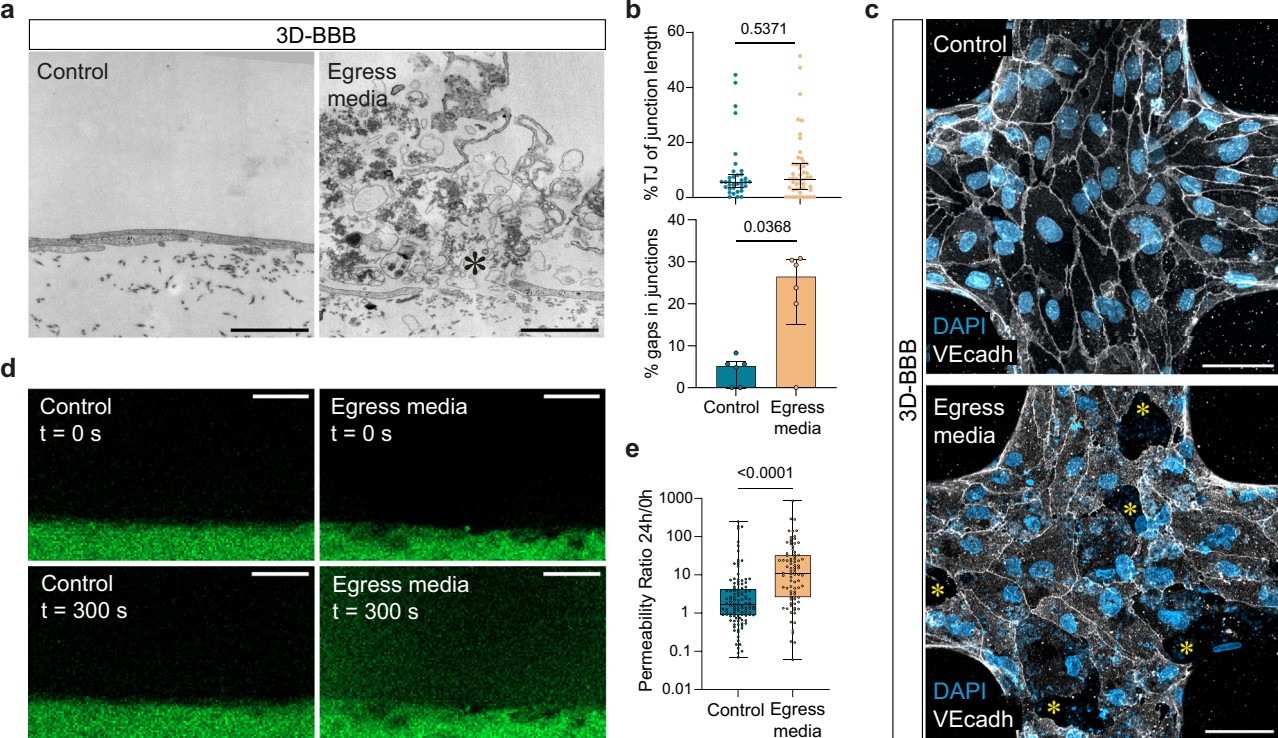

**Fig. 3 | *P. falciparum* iRBC-egress media causes inter-endothelial gaps impairing microvascular integrity. a** TEM images showing an inter-endothelial gap formed in 3D-BBB microvessels exposed to iRBC-egress media (right), compared to an intact junction in control microvessels (left). Asterisk indicates parasite egress products. Scale bar = 2 µm. **b** Scatter plot comparing the percentage of tight junctions over total junctional length in 3D-BBB microvessels exposed to control uRBC media (*N* = 1 microvessel) or iRBC-egress media (*N* = 2 microvessels) (top), and bar plot quantifying the percentage of inter-endothelial gaps found by TEM on 3D-BBB microvessels incubated with control uRBC media (*N* = 1) or iRBC-egress media (*N* = 2) (bottom). For both graphs, each dot represents the quantification of a TEM section from a different microvessel area with 10–25 quantified junctions, the median value is reported as a line with error bars indicating the IQR (Two-tailed Mann-Whitney U test). **c** Maximum z-projection of a confocal image showing the junctional marker VE-cadherin (white) in a microvessel incubated with control

media or iRBC-egress media. DAPI (blue) stains large nuclei corresponding to BBB cells, and nucleic acids found in iRBC-egress media, presenting small punctate labeling. Asterisks indicate inter-endothelial gaps. Scale bar = 50 µm. Representative image from at least 3 independent experiments with similar results. **d** Representative images showing 70 kDa FITC-dextran diffusion through the lateral wall of the 3D-BBB before and after 24 h incubation with control media or iRBC-egress media. Scale bar = 50 µm. **e** Ratio of the apparent permeability calculated at 24 h post-incubation and at pre-incubation with control media or iRBC-egress media. Each point represents the ratio from an ROI coming from 3D-BBB microvessels exposed to control media (*N* = 4 microvessels) and iRBC-egress media (*N* = 10) (Two-tailed Mann-Whitney U test). Box plots display the median (line within the box), IQR range (box), and range (whiskers). Data are provided as a Source Data file.

endothelial cells and regulation of cell migration in pericytes (Fig. 4b and Supplementary Data 2). *CellChat* analysis on upregulated transcripts revealed an increase in the expression of inflammatory ligand-receptor pairs following exposure to iRBC-egress media. Pericytes and astrocytes showed an increased expression of collagen-encoding genes, suggestive of a fibrotic, scar-forming phenotype (Fig. 4c and Supplementary Fig 3). In addition, all three cell types showed elevated expression of midkine (*MDK*), a chemoattractant for the recruitment of neutrophils, macrophages and lymphocytes[36].

Antigen presentation appeared to be an important inflammation-related process that was strongly elevated in all cell types that compose the 3D-BBB model. Notably, we found evidence of cellular uptake of parasite material by TEM. Endothelial cells in the 3D-BBB microvessels incubated with iRBC-egress media showed signs of activation, including the formation of membrane protrusions, large vacuoles containing iRBC membranes or hemozoin (Fig. 4d and Supplementary Fig. 4a), indicating the activation of parasite uptake. Immunofluorescent staining shows colocalization of parasite nucleic acids with lysosome-associated membrane protein 1 (LAMP1) (Fig. 4e), suggesting the delivery of parasite material to the lysosomal compartment of endothelial cells. We therefore defined transcriptomic gene signatures for antigen presentation (see Methods), either for major histocompatibility complex (MHC) class I transcripts or MHC class II transcripts (Fig. 4f). Endothelial cells and

astrocytes exhibited a robust upregulation of the MHC class I gene signature (effect size *r* > 0.5, *p* < 0.0001), while only a modest upregulation was observed in pericytes (effect size *r* = 0.2, *p* < 0.0001). Interestingly, the MHC class II antigen presentation gene signature, associated with CD4+ T-cell recruitment, was also strongly upregulated in endothelial cells (effect size *r* = 0.4, *p* < 0.0001) and modestly upregulated in astrocytes (effect size *r* = 0.2, *p* < 0.0001), with some astrocytes presenting particularly high MHC class II signature scores.

To gain deeper insights into other signaling pathways dysregulated upon exposure to *P. falciparum* egress products, we utilized PROGENy (Pathway RespOnsive GENes for activity inference)[37], a computational method that infers signaling pathway activities based on downstream gene expression. Consistent with the GO-term analysis, we observed an increase in NFκB and TNFα signaling in endothelial cells and pericytes upon challenge with iRBC-egress media. Notably, the JAK-STAT pathway was activated across all cell types (Fig. 4g), a result consistent with the upregulation of transcripts associated with the type I IFN response (Fig. 4a, b). To validate this result, we performed immunofluorescent staining of STAT1 on cell monolayers and 3D-BBB microvessels. Increased STAT1 expression was observed in the 3D-BBB model upon 24 h exposure to parasite egress products compared to the control condition, where the signal was barely detectable. In accordance with our scRNA-seq results, this increase occurred not only in endothelial

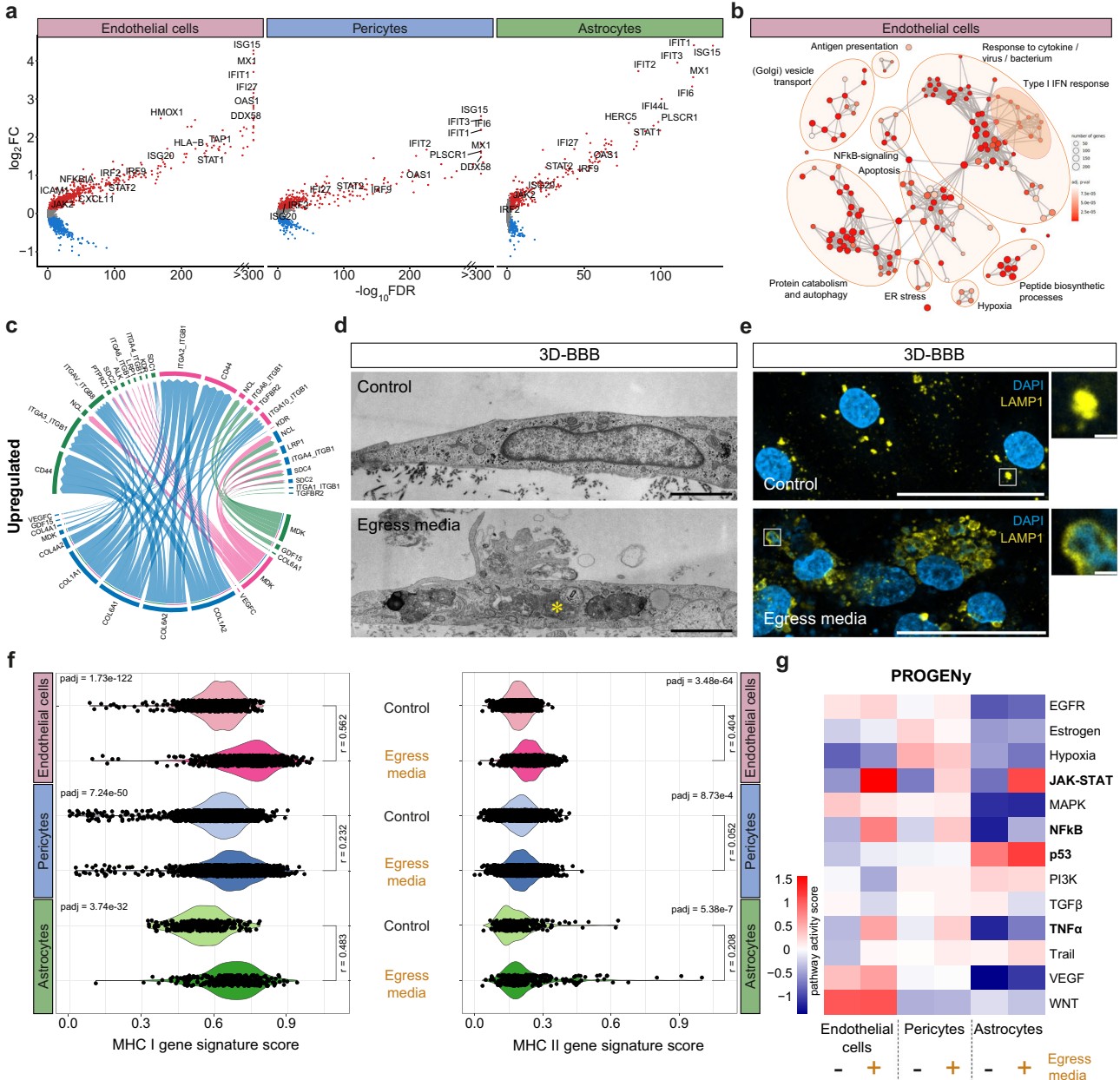

**Fig. 4 | iRBC-egress media activates inflammatory and antigen presentation pathways in all BBB cell types. a** Volcano plots of differentially expressed genes in endothelial cells (same analysis as in 3e), pericytes, and astrocytes upon 24-hour incubation with iRBC-egress media, plotting the log2-FC against the statistical significance (-log10 of the FDR). Significantly up- or downregulated genes (FDR < 0.05, log2FC > 0.1 or log2FC < − 0.1, respectively) are marked in red or blue and selected upregulated genes are labeled. **b** GO-term over-representation analysis on significantly upregulated genes (FDR < 0.05, log2FC > 0.1) in endothelial cells. Each network node represents one of the most significant GO-terms (adjusted *p*-value < 0.0001), and edges connect GO-terms with more than 20% gene overlap. GO-term clusters were manually summarized in one label term. *P*-values were calculated using the hypergeometric distribution (one-sided Fisher's exact test). Multiple comparisons adjustment was performed using the Benjamini-Hochberg procedure. **c** Selected upregulated ligand-receptor interactions identified among the cell types within the 3D-BBB model after exposure to iRBC-egress media using the *CellChat* package. Arrows point from colored ligands on sender cells to receptors on receiver cells. Weights of links are proportional to the interaction strength. **d** Representative TEM images of endothelial cells within the 3D-BBB model taking up parasite material (asterisk) after incubation with iRBC-egress media (bottom) and a control endothelial cell incubated with uRBC media (top). Scale bar = 2 μm. Additional TEM images can be found in Supplementary Fig. 4a. **e** Representative confocal images from two independent experiments (left) and close-up views (right) showing maximum z-projection of LAMP1 labeling (yellow) in 3D-BBB microvessels after 24-hour incubation with iRBC-egress media or control media. Scale bar = 50 μm; close-up = 2 μm. **f** Violin plots showing the MHC I and MHC II gene signature score plotted by cell type and condition (Two-tailed Mann-Whitney U test followed by calculation of effect size r, Benjamini-Hochberg multiple comparisons adjustment). Genes in the MHC I and II signature score can be found in Methods. **g** Heatmap of pathway activities inferred using the *PROGENy* method. The colors correspond to the mean PROGENy pathway activity score in the cells of the respective cell type and condition.

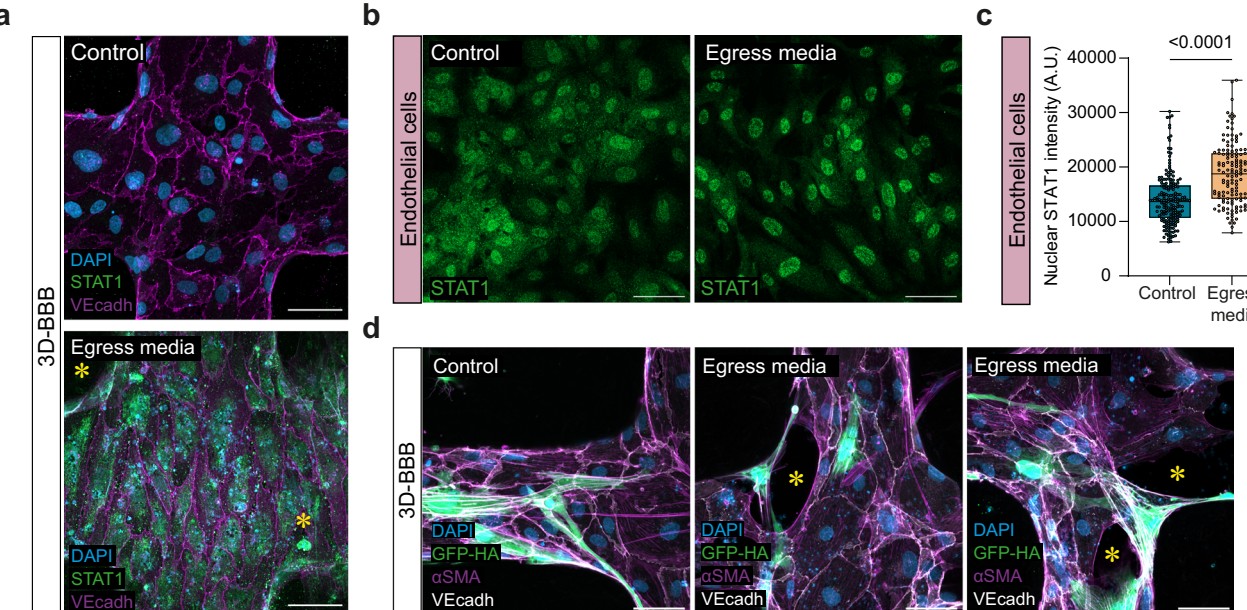

**Fig. 5 | *P. falciparum* iRBC-egress induces activation of the JAK-STAT pathway in all the BBB cells and induces changes in vessel architecture. a** Representative confocal imaging showing maximum z-projection of STAT1 labeling (green) in 3D-BBB microvessels after 24 h incubation with iRBC-egress media or control media. Asterisks indicate STAT1-positive astrocytes and pericytes in the collagen hydrogel. Endothelial junctions were labeled with VE-cadherin (magenta) and nuclei with DAPI (blue). Scale bar = 50 μm. **b** Representative maximum z-projection of confocal images showing STAT1 protein localization (green) in endothelial monolayers after 24 h incubation with iRBC-egress media or media control. Scale bar = 50 μm. **c** Mean fluorescence intensity of STAT1 labeling in the nuclei of endothelial cells grown on monolayers (*N* = 6 wells/condition) after 24 h incubation with iRBC-egress media or media control (Two-tailed Mann-Whitney U test). Box plots display the median (line), IQR (box), and range (whiskers). **d** Representative confocal imaging showing maximum z-projection of 3D-BBB microvessels after 24 h incubation with iRBC-egress media. GFP-expressing astrocytes (green) and αSMA-labeled pericytes (magenta), asterisks represent gaps between endothelial cells, stained with VE-cadherin (white). Scale bar = 50 μm. Representative images (**a** and **d**) are from at least 3 independent experiments with similar results. Source data are provided as a Source Data file.

cells, but also in the supporting pericytes and astrocytes present in the collagen hydrogel (Fig. 5a), indicating an overall response of the model to iRBC-egress media. STAT1 translocation to the nucleus was evaluated in 2D monolayers, as a proxy for increased pathway activity[38]. While endothelial cells presented increased STAT1 nuclear localization compared to the media-only condition, pericytes and astrocytes did not (Fig. 5b, c and Supplementary Fig. 4b, c). Furthermore, we found that the increase in the apoptosis-associated p53 pathway in astrocytes, as identified in the scRNA-seq analysis (Fig. 4g), was associated with a substantial reduction in cell density by immunofluorescence in astrocyte 2D monolayers (Supplementary Fig. 4d, e). This decrease occurred without an increase of astrocyte activation markers GFAP and ICAM-1 (Supplementary Fig. 4d). As a positive control for astrocytic activation, we treated astrocyte monolayers with TNFα, IL-1β and IFNγ or a cytokine cocktail at concentrations similar to those observed in CM patients[39], which resulted in an increase in both GFAP and ICAM-1 expression in the absence of a significant reduction in cell density (Supplementary Fig. 4d, e). Despite the lack of astrocyte reactivity after exposure to *P. falciparum*-egress media, astrocytes were often found to extend their processes towards regions of endothelial disruption in our 3D-BBB model (Fig. 5d). Taken together, our results suggest that *P. falciparum* products released upon egress cause a significant upregulation of inflammatory and antigen-presenting pathways, albeit with cell-specific differences.

### Binding of *P. falciparum*-iRBC for 6 h induces minor transcriptional changes

Next, we aimed to investigate whether blockade of endothelial receptors such as EPCR and ICAM-1 by iRBC binding directly contributes to BBB pathogenesis, given its strong association with CM[6,16,40]. We perfused 3D-BBB microvessels with highly synchronized trophozoite (26–34 h post infection) or schizont (38–46 h post infection) stages of *P. falciparum* HB3var03, a parasite line expressing a dual EPCR-ICAM-1 binding PfEMP1 (Supplementary Fig. 5a). Parasites or uninfected RBC were perfused at the same concentration as iRBC-egress media ($5 \times 10^7$ iRBC/mL) for 30 min, followed by a 20-minute wash to release unbound iRBC. Microvessels were incubated for 6 h, a time point that would prevent egress of parasites in the trophozoite condition, and analyzed morphologically through electron and confocal microscopy, as well as at the single-cell transcriptomic level (Fig. 6a). The UMAP confirmed the correct synchronization and development of *P. falciparum*-iRBC stages, as visualized by a trophozoite cluster positive for the *P. falciparum* mid-stage transcript *PfHB3_100020300* and a continuous, arch-shaped schizont cluster positive for late-stage marker *PfHB3_090035000* (Fig. 6b and Supplementary Fig. 5b). The scRNA-seq dataset was then filtered to exclude *P. falciparum*-iRBC from the analysis to focus on the transcriptional changes in the BBB cell types. We obtained 4514 quality-controlled cells, including all three conditions from the trophozoite, schizont and uninfected RBC microvessels. UMAP visualization after re-clustering of the BBB cells showed 4 distinct clusters, including an endothelial cluster (*CDH5* and *PECAM1*), an astrocyte cluster (*GFAP* and *S100B*) and two pericyte clusters (*PDGFRB*^high and *PDGFRB*^moderate) (Supplementary Fig. 5b–d). In contrast to exposure to *P. falciparum*-iRBC egress media, the UMAP representation revealed no clear segregation of cells based on the experimental conditions (Fig. 6c). Nevertheless, we identified some dysregulated transcripts in endothelial cells that were similar to those observed in cells treated with iRBC-egress media (Fig. 6d). Incubation with both trophozoite and schizont stages led to the downregulation of endothelial tight junction marker *CLDN5*, as well as of its regulator *SOX18*. Exposure to trophozoite stages caused an upregulation of genes encoding vesicle transport processes, including ER transcripts, such as *KDELR3*, or vesicular components, like *CAV1*, *COPB1* and *COPE* (Fig. 6d, e). Upon exposure to schizonts, we observed a

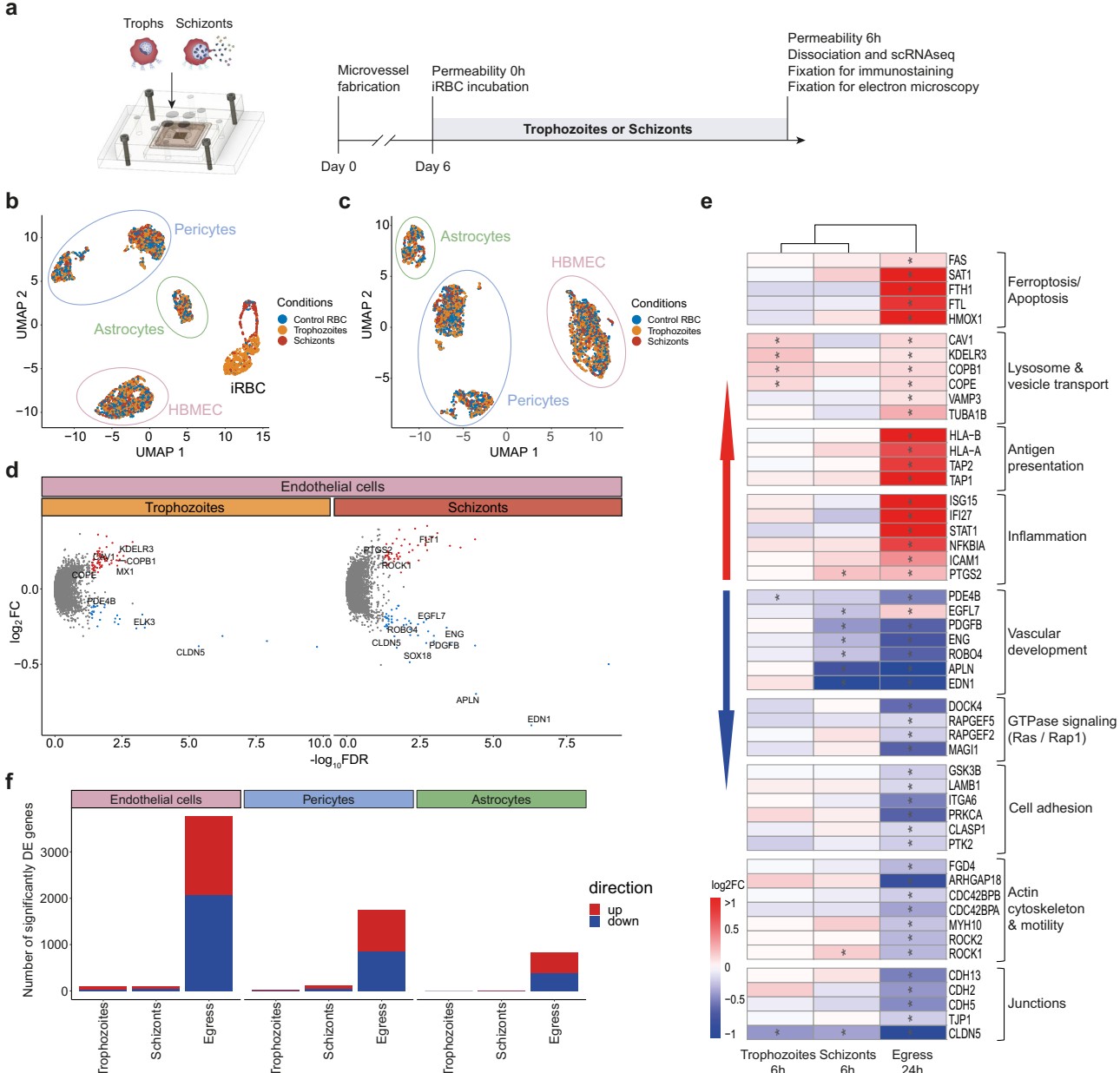

**Fig. 6 | Incubation with trophozoite- and schizont-stage *P. falciparum*-iRBC induces minor transcriptional changes in BBB cell types. a** Representation of the experimental timeline on 3D-BBB microvessels before and after 6-hour incubation with cytoadherent trophozoite or schizont *P. falciparum*-iRBC or control devices perfused with uRBC. **b** UMAP of sequenced BBB cells and *P. falciparum*-iRBC colored by experimental condition. **c** UMAP of BBB cells after exclusion of *P. falciparum*-iRBC. **d** Volcano plots of differentially expressed genes in endothelial cells upon 6-hour incubation with trophozoite- or schizont-stage iRBC, plotting the log2FC against the statistical significance (-log10 of the FDR). Significantly up- or downregulated genes (FDR < 0.05, log2FC > 0.1 or log2FC < − 0.1, respectively) are marked in red or blue, respectively, and selected genes are labeled. **e** Heatmap

showing log2FC values of selected genes belonging to significant GO-terms shown in Figs. 2f and 4b, in brain endothelial cells after 24-hour exposure to iRBC-egress media, or 6-hour exposure to trophozoite- or schizont-stage iRBC compared to the respective uRBC perfused control. A hierarchical clustering dendrogram was constructed based on the log2FC values of all genes differentially expressed in either of the experimental conditions. **f** Barplot showing the number of significantly differentially expressed genes (DE genes) (FDR < 0.05, log2FC > 0.1 or log2FC < − 0.1, respectively) in brain endothelial cells. Pericytes and astrocytes upon 3D-BBB microvessel exposure to trophozoite and schizont-stage iRBC, or iRBC-egress media, compared to the respective uRBC perfused controls.

downregulation of processes related to blood vessel development, including angiogenic, barrier formation, and endothelial migration-associated genes *APLN, END1, ENG, ROBO4* and *PDGFB* (Fig. 6e). Exposure to trophozoites and schizonts did not cause strong transcriptional changes in pericytes and astrocytes in the 3D-BBB model (Fig. 6f). Altogether, *P. falciparum*-iRBC causes minimal global transcriptional changes in human cells present in the 3D-BBB model.

## The egress of *P. falciparum*-iRBC locally increases barrier permeability and disrupts junctional morphology

Despite the lack of a major transcriptional shift, some of the dysregulated transcripts in 3D-BBB microvessels exposed to trophozoite and schizont-stage iRBC were suggestive of a potential barrier dysfunction. Although TEM did not reveal any changes in the percentage of total junction length covered by tight junctions among the three

examined conditions (Fig. 7a, b), we observed junctional differences by immunofluorescence staining. VE-cadherin labeling revealed the presence of thin junctions and inter-endothelial gaps highly localized in microvessel regions near egressed merozoites, which could be identified as small punctate signal (<1 μm) by DAPI staining (Fig. 7c). Specifically, the presence of inter-endothelial gaps was minimal in 3D-BBB microvessel regions exposed to high *P. falciparum*-iRBC cytoadhesion with low egress (i.e., where iRBC looked intact and free merozoite were barely present), with 5 (IQR = 4–6) gaps per field of view, compared to 27 (IQR = 19.5–38) gaps in microvessels with high rate of schizont rupture, largely colocalizing with egressed merozoites (Fig. 7d). Even though no visible changes on the length of tight junctions were observed by TEM, we identified extravasated merozoites in the collagen matrix of 3D-BBB microvessels incubated with schizonts (Fig. 7e). Furthermore, *P. falciparum* schizonts induced functional alterations in barrier integrity, with a 3-fold increase in permeability to 70 kDa FITC-dextran upon 6-hour incubation with schizonts, with median permeability ratios of 0.88 (IQR = 0.51, 1.72) in control and 2.46 (IQR = 1.54, 6.77) in 3D-BBB microvessels exposed to schizonts (Fig. 7f). Collectively, these data suggest that although the effects of *P. falciparum*-iRBC sequestration on endothelial cells are local and associated to the egress of parasite components, they still result in changes in vessel permeability.

### The egress of *P. falciparum*-iRBC causes localized transcriptional changes

To determine if the local disruption of endothelial cells was accompanied by a transcriptional shift, we defined a *P. falciparum*-iRBC *egress signature score*, including the 50 most upregulated and downregulated genes in endothelial cells exposed to iRBC-egress media (see Methods and Supplementary Data 1). Endothelial cells exposed to both trophozoite or schizont *P. falciparum*-iRBC presented a significant increase in the *egress signature score*, which was higher in cells exposed to schizonts (Fig. 7g). Interestingly, the *egress signature score* of endothelial cells exposed to schizont *P. falciparum*-iRBC showed a bimodal distribution compared to the unimodal distribution in the two remaining conditions (Fig. 7g and Supplementary Fig. 5e), suggestive of two transcriptional endothelial states. To infer if this bimodality might be related to spatial proximity to regions of high *P. falciparum*-iRBC egress, we defined an endothelial population that contained *P. falciparum* gene counts above an estimated background-level threshold (see Methods), indicating uptake of parasite material by the respective endothelial cells. Indeed, this population presented a significant increase in the *egress signature score*, compared to cells with a minimal background level of *P. falciparum* reads (*p*-value = 0.0086) (Fig. 7h). Furthermore, a deeper analysis of the transcriptional alterations in this *P. falciparum*-RNA-high cell population revealed a remarkably similar transcriptional profile to the one of endothelial cells exposed to egress products (Fig. 7i). This included a strong downregulation of transcripts associated with actin cytoskeleton and cell motility, focal adhesions, RAS/RAP1 GTPase pathway, along with low expression of genes encoding junctional and vascular development pathways. Although only a minor increase in antigen presentation and inflammation pathways was observed compared to iRBC-egress media, an equally strong upregulation of ferroptosis and vesicle transport genes was found (Fig. 7i). Taken together, these results suggest that egress of malaria components from *P. falciparum*-iRBC causes a strong, localized and well-defined signature in endothelial transcription, similar to that found globally in microvessels exposed to *P. falciparum* egress products.

### Pharmacological inhibition of JAK-STAT signaling preserves 3D-BBB integrity

To investigate the relevance of the JAK-STAT pathway as a mediator of the vascular dysfunction caused by *P. falciparum* egress products, and to evaluate its potential as a therapeutic target, we treated our 3D-BBB microvessels with Ruxolitinib, a clinically approved JAK1/2 inhibitor. Briefly, microvessels were incubated for 24 h with either iRBC-egress media or control cell culture media, in the presence or absence of 10 μM Ruxolitinib. Immunofluorescence staining confirmed that Ruxolitinib prevented the increase in STAT-1 expression induced by *P. falciparum*-egress media (Fig. 8a). Furthermore, the presence of the drug preserved the junctional localization of the tight junction protein ZO-1 in microvessels exposed to *P. falciparum* egress products (Fig. 8b). To determine whether these changes were associated with an improvement in 3D-BBB barrier function, microvascular permeability was assessed by 70 kDa FITC-dextran perfusion before and after treatment, as described in previous sections. While treatment with Ruxolitinib modestly improved baseline permeability in control devices (0.66 − IQR = 0.26, 1.51), its impact was far more pronounced under pathological conditions. Co-incubation with iRBC-egress media led to an almost 24-fold reduction in microvascular permeability (0.45 − IQR = 0.10, 1.94) compared to the iRBC-egress media condition alone (Fig. 8c). Overall, these results highlight the importance of the JAK-STAT pathway in *P. falciparum*-mediated barrier disruption and support the therapeutic potential of Ruxolitinib in reducing brain vascular dysfunction associated with cerebral malaria.

## Discussion

CM is characterized by *P. falciparum* accumulation in the brain microvasculature, and it is often accompanied by vascular dysfunction[2,3]. Despite the severity of the disease, our understanding of how the malaria parasite affects the BBB remains limited, primarily due to difficulties in obtaining brain samples from affected patients or the lack of accurate disease models. In this study, we developed a bioengineered microvascular 3D-BBB model that incorporates primary human brain microvascular endothelial cells, pericytes and astrocytes. The addition of these cell types increased the vascular barrier function of the model, improving upon other bioengineered models previously used to study *P. falciparum* pathogenesis[24,30,31]. We used this advanced model to assess BBB disruptive mechanisms mediated by *P. falciparum*. Perfusion and incubation either with media containing *P. falciparum*-egress products or with *P. falciparum* schizonts led to a global increase in microvascular permeability to 70 kDa FITC-dextran. However, differences in the extent of vascular barrier opening were found between these two conditions. While large VE-cadherin inter-endothelial gaps formed after incubation with iRBC-egress media, schizont sequestration only induced smaller gaps near areas of high merozoite egress. Similar differences in the extent and ubiquity of parasite-induced transcriptional shifts were found in our scRNA-seq analysis. We observed major and widespread alterations in endothelial gene expression after exposure to iRBC-egress media, and only minimal global differential expression after 6 h incubation with trophozoites and schizonts. This analysis is in agreement with other bulk transcriptomics studies that showed limited endothelial transcriptional changes upon incubation with *P. falciparum*-iRBC[30] and more prominent changes after incubation with *P. falciparum* lysates[29]. Nevertheless, the single-cell resolution of our transcriptomic analysis revealed that endothelial cells near regions of egress, identified by high *P. falciparum* RNA content, presented a transcriptional signature similar to that of endothelial cells exposed to iRBC-egress media. Altogether, our results show that although *P. falciparum*-mediated disruption is highly localized to areas of *P. falciparum* egress, it could still result in severe pathogenic outcomes, as shown by the increase in barrier permeability we observed.

Although *P. falciparum* has a broad repertoire of members of the parasite ligand PfEMP1, CM patients are enriched in variants that bind to EPCR[41]. Whether parasite binding to specific endothelial receptors directly contributes to vascular disruption remains unknown. Recent studies have reported conflicting results on the transcriptional effect

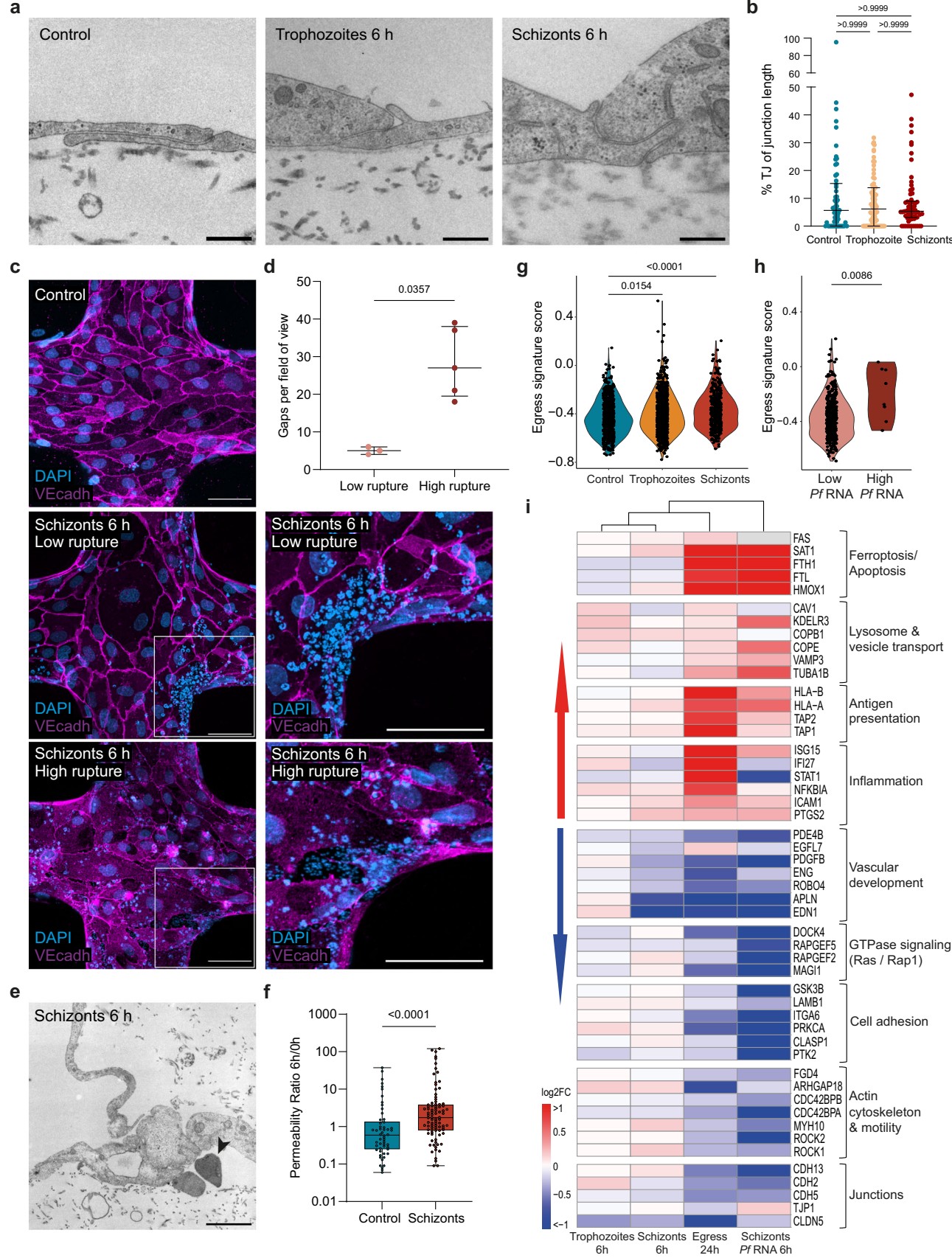

**Fig. 7 | Incubation with *P. falciparum*-iRBC induces localized changes at sites of parasite egress. a** Representative TEM images showing inter-endothelial junctions formed in 3D-BBB microvessels exposed to trophozoites, schizonts or uRBC. Scale bar = 500 nm. **b** Scatter plot of percentage of tight junctions per total junctional length after incubation with control media, trophozoites and schizonts (N = 3 microvessels/ condition) (Kruskal-Wallis with Dunn's multiple comparisons test). Each dot represents images from TEM sections, including 58–74 junctions, the lines represent the median and error bars indicate IQR. **c** Representative z-projection confocal images (left) and close-up (right) of VE-cadherin in control microvessel and after 6 h incubation with schizonts. DAPI identifies mature schizonts (~5 μm) and merozoites (<1 μm). Scale bar = 50 μm. **d** Scatter plot quantifying the colocalization of gaps and merozoites in ROI of microvessels with low (N = 3) or high (N = 5) schizont rupture. The lines represent the median, and error bars indicate IQR. **e** TEM image showing an extravasated merozoite (arrowhead) in the collagen of 3D-BBB exposed to schizonts. Scale bar = 2 μm. **f** Ratio of apparent permeability between values at 6-hour post-incubation with control or schizonts and baseline permeability before incubation. Each point represents a different ROI from 3D-BBB microvessels treated with schizonts (N = 8) or control media (N = 5) (Two-tailed Mann-Whitney U test). Box plots display the median (line), IQR (box), and range (whiskers). **g** Violin plot of *egress signature score* in every HBMEC across conditions (Kruskal-Wallis with Dunn's multiple comparisons test). **h** Violin plot of *egress signature score* in HBMEC exposed to schizonts split by *P. falciparum* RNA content (high > 10 logcounts) (*Pf* RNA low: N = 385, *Pf* RNA high: N = 8) (Two-tailed Mann-Whitney U test). **i** Heatmap showing log2FC values of selected endothelial genes in Figs. 2f and 4b, after exposure to iRBC-egress media, trophozoites, schizonts, or endothelial cells with high *P. falciparum*-RNA content compared to uRBC control. A hierarchical clustering dendrogram was constructed based on the log2FC values of all genes differentially expressed in either of the experimental conditions. Data are provided as a Source Data file.

of cytoadherent *P. falciparum*-iRBC. Studies on endothelial-only 3D brain microvessels have shown minimal transcriptional differences after short-term incubation with *P. falciparum*-iRBC[30]. Likewise, no differences in key endothelial transcripts were found between parasites derived from CM and uncomplicated malaria patients in a study in Malawi[42]. Conversely, a recent study has shown differential endothelial gene expression upon binding of parasite lines expressing different PfEMP1[43]. Our study did not find major transcriptional differences after short term incubation with highly synchronized trophozoites expressing a dual EPCR-ICAM-1 binding PfEMP1 previously shown to be highly disruptive in in vitro BBB spheroids[14]. Overall, the lack of transcriptional differences could be a result of the short incubation period, chosen to disentangle the effects of binding from that of natural parasite egress. We cannot rule out the possibility of a *P. falciparum* binding-induced transcriptional shift at later time points, and the occurrence of non-transcriptional cellular processes not evaluated in our study, including morphological or mechanical modifications on endothelial cells, such as transmigratory-like cup structures or trogocytosis, previously identified in *P. falciparum* in vitro studies[44]. Other important disruptive effects not investigated in this study are synergistic damaging effects in the presence of other co-factors, such as protein C and thrombin[30,45].

Our study confirms that the egress of sequestered *P. falciparum* parasites in close proximity to endothelial cells is a key pathogenic event, accompanied by the release of agents such as heme[17], hemozoin[18], parasite histones[19,20], PfHRP2[21], and GPI anchors[23], which have been previously described to activate endothelial cells and compromise barrier integrity. We reproduced *P. falciparum*-induced transcriptomic signatures found in other studies[29,30], including those related to endothelial disruption of ER-transport, oxidative stress and ferroptosis. These changes likely resulted from the detoxification of hemozoin, heme and other parasite products[46]. Nevertheless, the use of an improved 3D-BBB model revealed new parasite-disruptive mechanisms. We showed for the first time that *P. falciparum* induces a global endothelial downregulation of the tight junction marker *CLDN5* (Claudin-5) in all the parasite conditions we tested, as well as the decrease of other junctional transcripts in microvessels treated with iRBC-egress media (Fig. 8d). Of additional relevance is the decrease in expression of vascular developmental genes, as well as in transcripts from pathways related to cell adhesion and cytoskeletal organization. Our study also shows the downregulation of important homeostatic signaling pathways between endothelial cells and pericytes[47], including the PDGF-PDGFR[48] and Notch signaling pathways[49], and the Angiopoietin/Tie-2 axis[31]. These data are in agreement with a recent study from our group demonstrating that pericytes play a functional role in CM pathogenesis by halting their expression of Ang-1[31]. Upregulated pathways in astrocytes and pericytes in response to iRBC-egress media include collagen secretion, suggestive of fibrosis, and p53-mediated astrocytic apoptosis, which could explain the reduced

astrocyte density observed by immunofluorescence staining. However, we did not observe any increase in astrocyte activation and GFAP expression in response to *P. falciparum*. Post-mortem studies on CM samples have revealed different degrees of astrogliosis, although not co-localizing with iRBC sequestration sites[3], suggesting that astrocyte reactivity may be caused by a different pathogenic driver.

Our study has revealed that all the cell types included in the 3D-BBB model present upregulation of type I IFN response and antigen presentation pathways. Interestingly, polymorphisms of the IFN-alpha receptor-1 (*IFNAR1*)[50] or *IFIT1*[51] have been associated with a reduced risk of CM. Our findings align with previous reports of type I IFN response in a *P. berghei* experimental cerebral malaria model[52]. Other studies on *P. berghei* CM models have suggested a potential mechanism of merozoite engulfment and cross-presentation of parasite antigens by endothelial cells[34,35], or by astrocytes and microglia[53]. Our model suggests the existence of similar engulfment mechanisms of *P. falciparum* egress products, likely responsible for the activation of antigen presentation pathways in endothelial cells, but also in pericytes and astrocytes within the collagen hydrogel. Our findings and others[44,52,53] suggest a potential mechanism for leukocyte recruitment not only intravascularly by endothelial cells[30], but also at the brain perivascular space, aligning with recent observations of vascular and perivascular accumulation of CD8+ T-cells in the brain microvasculature in post-mortem samples of CM patients[54,55]. In this scenario, our results suggest that BBB cells might be able to present internalized parasite material to CD8+ T-cells, which could potentially amplify damage at the vascular and perivascular space.

Another inflammation-associated response highly upregulated in the three BBB cell types was the JAK-STAT pathway, as indicated by the strong transcriptional activation of members of this pathway, as well as by the nuclear translocation of STAT1 in endothelial cells after incubation with iRBC-egress media. Enhancement of leukocyte activation of type I and II IFN and pro-inflammatory pathways, such as JAK-STAT, is a prominent systemic feature in *P. falciparum* infection (reviewed in ref. 56). Notably, to explore the therapeutic potential of targeting this pathway, we treated 3D-BBB microvessels with Ruxolitinib, a clinically approved JAK1/2 inhibitor. Our data demonstrate that Ruxolitinib effectively prevented vascular disruption induced by *P. falciparum* egress products, as evidenced by preserved junctional integrity and reduced microvascular permeability (Fig. 8d). Interestingly, Ruxolitinib also conferred a moderate protective effect in the absence of *P. falciparum*-associated stimuli, suggesting the dual capacity of JAK-STAT inhibition to stabilize endothelial function both under inflammatory and resting conditions. Recent studies in clinical human malaria infections have shown the tolerability of Ruxolitinib in combination with antimalarials[57], and its efficacy as an adjunctive therapy in uncomplicated infections by attenuating inflammation and endothelial activation early in infection[58], without affecting immune memory responses[58,59]. Altogether, our results, combined with emerging data in human infected

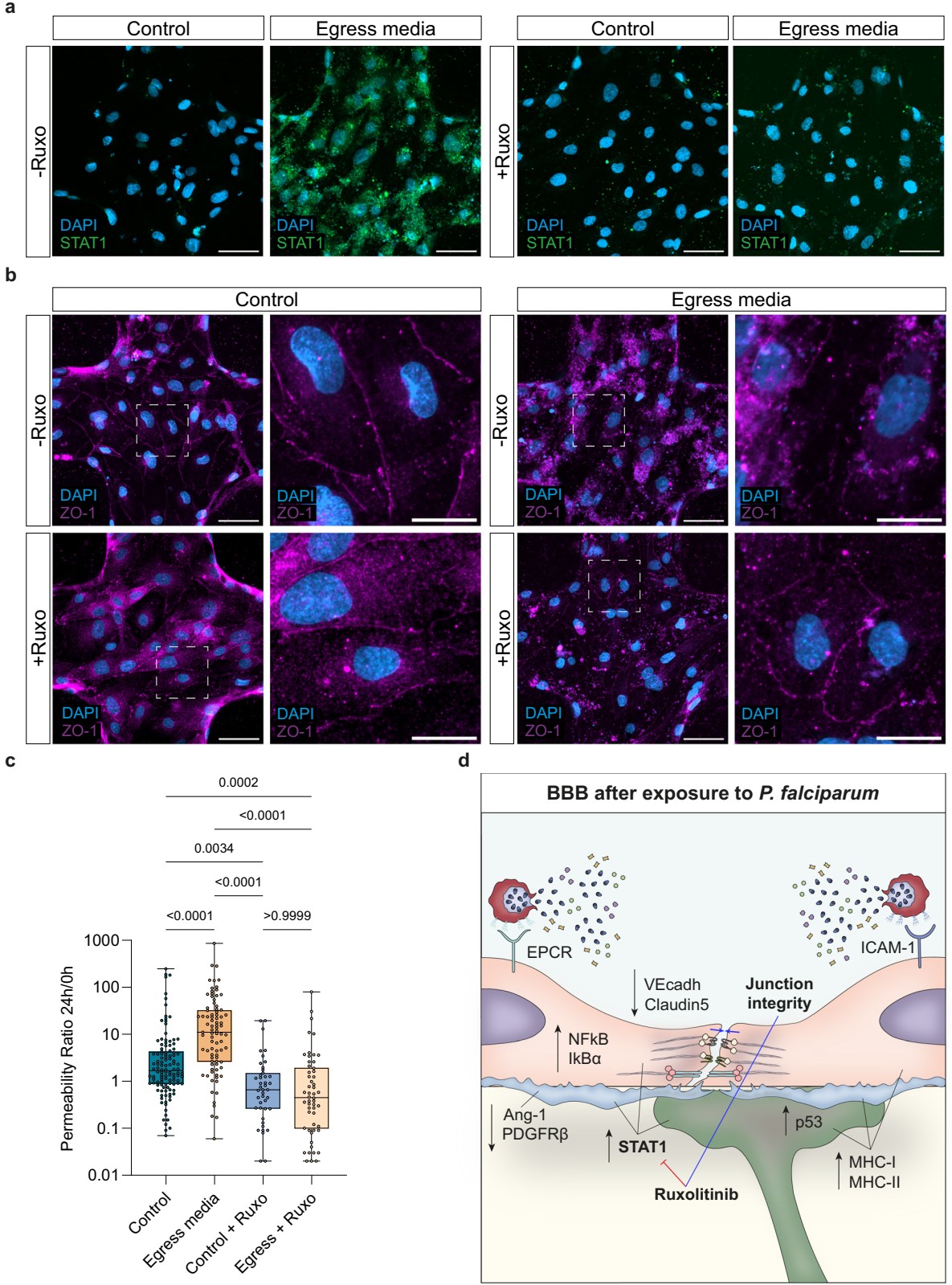

**Fig. 8 | Pharmacological inhibition of JAK-STAT signaling preserves 3D-BBB integrity. a** Representative maximum z-projection confocal images of STAT1 protein (green) in 3D-BBB microvessels in control microvessels and after 24 h incubation with iRBC-egress media, and upon co-incubation with Ruxolitinib. Scale bar = 50 μm. **b** Representative maximum z-projection confocal images (left; scale bar = 50 μm) and close-up views (right; scale bar = 20 μm) of junctional marker ZO-1 (magenta) in control microvessels and after 24 h incubation with iRBC-egress media, and upon co-incubation with Ruxolitinib. **c** Ratio of the apparent permeability between values calculated at 24 h post-incubation with control media or iRBC-egress media, and at pre-incubation. Each point represents the ratio from an ROI from 3D-BBB microvessels exposed to control media ($N = 4$), iRBC-egress media ($N = 10$) or co-incubated with Ruxolitinib and control media ($N = 3$) or iRBC-egress media ($N = 4$) (Kruskal-Wallis with Dunn's multiple comparisons test). Box plots display the median (line), IQR (box), and range (whiskers). **d** Schematic model of the altered pathways in the BBB cell types upon iRBC-egress media exposure and at sites of natural iRBC rupture. *P. falciparum* egress products released upon iRBC rupture activate antigen presentation (MHC I, MHC II) and inflammation-related (STAT1) pathways in BBB cell types. Furthermore, while endothelial cells down-regulate junctional markers, pericytes show reduced Ang-1 and PDGFRβ signaling and astrocytes activate the p53 apoptosis-associated pathway. Pharmacological inhibition of the JAK-STAT pathway through Ruxolitinib prevents STAT1 over-expression and preserves vascular barrier integrity. Representative images in (**a** and **b**) are from 2 independent experiments with similar results. Data are provided as a Source Data file.

volunteers, suggest that inhibitors of the JAK-STAT pathway, such as Ruxolitinib, could be repurposed not only to modulate systemic immune responses but also to directly protect vascular integrity and mitigate cerebral complications during *P. falciparum* infection.

While our 3D-BBB model represents a significant advancement over previous in vitro systems, it still presents several limitations. Microvessel fabrication is complex and requires a long period of training in experienced laboratories. Although presenting improved barrier properties, our model is still far away from recreating BBB physiological permeability rates, which is likely related to the inherently low expression of tight junction proteins like Claudin-5 and Occludin in primary endothelial cells[60,61]. While the barrier properties of our 3D-BBB microvessels could be enhanced using induced pluripotent stem cell (iPSC)-derived brain endothelial cells[62], concerns have been raised about their epithelial-like characteristics[63], particularly in early protocols. Recent advances have shown that over-expression of ETS factors, such as ETV2, FLI1, ERG[63], or FOXF2 and ZIC3[64] can shift these cells towards an improved vascular and brain vascular phenotype, respectively. As other differentiation mechanisms are currently being developed for the generation of iPSC-derived BBB models[65,66], iPSCs still remain a promising tool for generating in vitro models of the microvasculature of the brain and other organs affected during *P. falciparum* infection[67]. Another limitation of our study is that we have solely focused on the pathogenic effect of *P. falciparum* and have not evaluated the effect of cytokines[39] or immune cell types[54,68]. Future studies could take advantage of the controlled microfluidic properties of our model to introduce these components, which play a relevant role in CM. Finally, although our 3D-BBB model presents increased complexity, other brain cell types, including microglia and neurons, could be incorporated to further resemble human physiology and investigate brain-associated pathogenic mechanisms. Despite these limitations, our model has shed light on novel pathogenic pathways of *P. falciparum* malaria and highlights the value of using innovative bioengineered models to enhance our understanding of infection and facilitate the development of future treatments.

## Methods
### Primary human cell culture
Primary human brain microvascular endothelial cells (HBMEC, Lot 376.05.04.01.2 F or 376.11.04.01.2 F, Cell Systems) were cultured in endothelial cell growth medium-microvascular (EGM-2MV, Lonza) up to passage 8. Primary human astrocytes (HA, Lot 31978, ScienCell) were cultured in basal media supplemented with 2% FBS, 1% Pen-Strep solution and 1% astrocyte growth supplement (ScienCell) up to passage 6. Primary human brain vascular pericytes (HBVP, Lot 32562, ScienCell) were cultured in basal media supplemented with 2% FBS, 1% Pen-Strep solution and 1% pericyte growth supplement (ScienCell) up to passage 8. All cell types were cultured at 37 °C and 5% CO2 as monolayers until microvessel fabrication, using a poly-L-lysine (0.1% (w/v), Sigma-Aldrich) coating for cell attachment.

### Green fluorescent protein and mCherry lentiviral transduction
GFP-expressing HA and mCherry-expressing HBVP were obtained by lentiviral transduction. Briefly, cells were grown in a T75 flask and, once confluent, they were incubated for 24 h in serum-free astrocyte or pericyte media with concentrated viral particles containing a GFP or mCherry vector (kindly donated by the laboratory of Dr Kristina Haase, EMBL Barcelona), at a multiplicity of infection of 10. After 24 h, the lentiviral particles were removed with two consecutive 24-hour washes in their respective cell culture media before astrocytes and pericytes were expanded and frozen down. For GPF-expressing HA, the efficiency of transduction was quantified as a percentage of GFP signal overlapping with the cell-covered area. mCherry-positive pericytes were further selected by fluorescence-activated cell sorting.

### *P. falciparum* culture
HB3var03 *P. falciparum* parasites were cultured using human 0+ erythrocytes in RPMI 1640 medium (GIBCO) supplemented with 10% human type-AB+ plasma, 5 mM glucose, 0.4 mM hypoxantine, 26.8 mM sodium bicarbonate and 1.5 g/L gentamicin (RPMI complete). Parasites were grown in a gas mixture of 90% N2, 5% CO2, and 5% O2, and parasitemia was regularly checked by Giemsa staining to avoid culture overgrowth. Cultures were regularly panned and monitored for correct PfEMP1 expression (see *Quantitative Polymerase Chain Reaction* section). *P. falciparum* parasites were synchronized weekly using 5% sorbitol to select for ring-stage parasites and 70% Percoll gradient to select for schizonts.

### 3D-BBB microvessel fabrication
The protocol for 3D microvessel fabrication can be found in ref. 25, and we conducted the following modifications for the generation of a 3D-BBB model. Briefly, type I collagen was isolated from rat tails as previously described, and dissolved in 0.1% acetic acid to a stock concentration of 15 mg/mL before dilution to 7.5 mg/mL and neutralized on EGM-2MV supplemented with 1% astrocyte and pericyte growth factors (ScienCell) before HBMEC seeding for microvessel fabrication. Primary HA and HBVP were added to the neutralized collagen solution in a 7:3 ratio, using a concentration of $7.5 \times 10^5$ HA/mL(collagen): $3.2 \times 10^5$ HBVP/mL(collagen). A 1:1 astrocyte-to-pericyte ratio ($2.5 \times 10^5$ HA/mL(collagen): $2.5 \times 10^5$ HBVP/mL(collagen)) was initially used to compare the expression level of BBB-specific markers by quantitative polymerase chain reaction. A multi-step process combining soft lithography and injection molding was used to fabricate microvessels, as previously reported[25,69]. Briefly, the top and bottom parts of the microvessels were fabricated separately and then assembled within two polymeric housing jigs. The negative impression of a microfluidic network was obtained by injecting the collagen solution between the top polymeric jig and a positive polydimethylsiloxane (PDMS) micro-patterned mold, previously made hydrophilic by O2 plasma treatment. The bottom part was fabricated by pouring the collagen solution on top of a 22 mm × 22 mm coverslip within the bottom housing jig and by

compressing it using a flat PDMS stamp to obtain a thin collagen layer. The two pieces were left gelling up to 1 h at 37 °C and then assembled after removal of the PDMS molds. The microvessels were incubated for at least one hour with EGM-2MV medium supplemented with 1% astrocyte and pericyte growth factors (ScienCell) before HBMEC seeding. Primary HBMEC were seeded at a concentration of $7 \times 10^6$ cells/mL under gravity-driven flow by adding 8 μL volume increments to the device inlet until reaching full coverage in the microfluidic network. Microvessels were cultured for up to 7 days and fed every 12 h by gravity-driven flow.

### Quantitative polymerase chain reaction on 3D-BBB microvessels and *P. falciparum*

After microvessels fixation at 0.5, 3 or 7 days in culture, total RNA was isolated from disassembled collagen hydrogels using TRIzol LS and then purified by RNeasy Micro Kit (Qiagen 50974004). The purified RNA was quantified using a NanoDrop™ 2000c Spectrophotometer (ThermoFisher) and then converted to the complementary cDNA using the TaqMan Reverse Transcription Reagents (ThermoFisher, N8080234) according to the manufacturer's instructions. Quantitative polymerase chain reaction (qPCR) was performed using the Light-Cycler 480 SYBR Green I Master (Roche, 04707516001) in a Light-Cycler 480 II (Roche). The oligonucleotides used as primer sequences for this qPCR experiment were purchased from ThermoFisher and are reported in Supplementary Data 3. The PCR program consists of an initial step at 95 °C for 15 min, followed by 45 cycles of 30-second denaturation at 94 °C, 40 s annealing at 60 °C and 50 s extension at 72 °C. The automatically detected threshold cycle values were compared using the ΔΔCt method, with the 0.5-day condition as the reference for comparison, and the gene expression levels were normalized to those of the housekeeping gene *PECAM1*.

qPCR was used to monitor PfEMP1 variant expression of *P. falciparum* line HB3var03. Parasites were synchronized with 5% sorbitol, and early ring stage parasite RNA (0-22 h post-infection - hpi) was extracted with TRIzol LS and RNeasy Micro Kit, and reverse transcribed with TaqMan™ Reverse Transcription Reagents as described above. The expression of the var repertoire in *P. falciparum* is monitored with an HB3 primer repertoire reported in Supplementary Data 3, using LightCycler 480 SYBR Green I Master (Roche, 04707516001) in a LightCycler 480 II (Roche). The following PCR conditions were used: 50 °C for 1 min, 95 °C for 10 min, then 40 cycles of dissociation, annealing, and extension at 95 °C for 15 sec, 52 °C for 15 sec, and 60 °C for 45 sec, respectively. Relative transcription of var genes was normalized to the housekeeping control seryl-tRNA-synthetase (STS; PF07_0073), and the expression was represented as relative gene expression $2^{(-\Delta CT)}$ (Supplementary Fig. 5a).

### Immunofluorescent staining of 2D monolayers

HBMEC, HA or HBVP were seeded on poly-L-lysine-coated 8-well slides (Falcon) at a concentration of $2 \times 10^4$ cells/well and grown until reaching confluency. 2D monolayers were fixed for 20 min with ice-cold 4% PFA. Fixation was followed by two consecutive phosphate-buffered saline (PBS) washes and a 1 h blocking-permeabilization solution in 2% BSA and 0.1% Triton X-100 in PBS at room temperature. Primary antibodies were diluted in a 2% BSA and 0.1% Triton X-100 PBS solution and incubated for 1 h at room temperature. Primary antibodies against the following proteins were used: VE-cadherin (Santa Cruz Biotechnology sc-52751 or Abcam ab33168, 1:100), STAT1 (Cell Signaling 14994S, 1:100), vWF (Bio-Rad AHP062, 1:200), PECAM1 (BD Pharmingen 560983, 1:100), β-catenin (Santa Cruz Biotechnology sc-59737, 1:200), ICAM-1 (Abcam ab20, 1:200), GFAP (Abcam ab4674, 1:200), S100B (Sigma S2532-100U, 1:200), AQP4 (Novus Biologicals NBP1-87679, 1:200), αSMA (Abcam ab202509, 1:200), PDGFRβ (Abcam ab69506, 1:200), NG2 (Invitrogen 372700,

1:200). After two PBS washes, the monolayers were incubated with 2 μg/mL DAPI (ThermoFisher D21490, 1:250), Alexa-Fluor 488-, Alexa-Fluor 594- or Alexa-Fluor 647-conjugated secondary antibodies (Invitrogen, 1:250) for 1 h at room temperature and then washed twice with PBS. Images were acquired using an LSM980 Airyscan 2 microscope (Zeiss) and processed with imaging software ZEN (Zeiss, v3.3.89) and Fiji (ImageJ, v1.54 f).

### Immunofluorescent staining of 3D microvessels

3D-BBB microvessels were fixed with ice-cold 4% paraformaldehyde (PFA) for 20 min after 7 days in culture. All the solutions for 3D microvessel staining were perfused through gravity-driven flow. Fixation was followed by two consecutive 10-minute PBS washes before immunofluorescent staining. Any possible background signal coming from the collagen hydrogel was quenched using Background Buster (Innovex Biosciences) for 30 min. After blocking/permeabilization in 2% BSA and 0.1% Triton X-100 (in PBS) at room temperature for 1 h, the devices were incubated at 4 °C overnight in a PBS solution containing the primary antibodies, 2% BSA and 0.1% Triton X-100. Primary antibodies against the following proteins were used: GFP (Invitrogen A21311, 1:200), mCherry (Invitrogen M11240, 1:200), VE-cadherin (Santa Cruz Biotechnology sc-52751, 1:100 or Abcam ab33168, 1:100), STAT1 (Cell Signaling 14994S, 1:100), αSMA (Abcam ab202509, 1:200), LAMP1 (Cell Signaling 9091, 1:100), ZO-1 (Invitrogen 339100, 1:100). After six 10 min PBS washes, the microvessels were incubated with 2 μg/mL DAPI (ThermoFisher D21490, 1:250), Alexa-Fluor 488-, Alexa-Fluor 594- or Alexa-Fluor 647-conjugated secondary antibodies (Invitrogen, 1:250) for 1 h at room temperature and then washed six times for 10 min with PBS. Imaging of the devices was performed using an LSM980 Airyscan 2 microscope (Zeiss), and images were processed with imaging software ZEN (Zeiss, v3.3.89), Fiji (ImageJ, v1.54 f) and Vision4D (Arivis, v3.5.1).

### Confocal image analysis

Confocal images were analyzed using Fiji (ImageJ, v1.54 f) for quantification of different parameters. For each image, Z-stack slices were summed to preserve all the fluorescent signal and the images were split into different channels and analyzed as 2D images. Quantification of gaps at sites of *P. falciparum*-iRBC rupture was obtained manually counting the number of gaps that colocalized with at least one free merozoite. The gap number per field of view was expressed as mean ± standard deviation. For nuclear STAT1 quantification, the DAPI channel was used to create a mask of the nuclei to select the regions of interest (ROIs). For each ROI, we measured the mean fluorescent value in the STAT1 channel, and we then compared the median nuclear fluorescent intensity among conditions. To obtain astrocyte density, the DAPI channel was used to create a mask and count the nuclei per field of view.

### *P. falciparum*-iRBC egress media preparation

The protocol for the generation and purification of *P. falciparum*-iRBC can be found in ref. 31. Briefly, *P. falciparum*-iRBC at 42–48 hpi were purified by a gelaspan (40 mg/mL) gradient separation and treated for 5 h with the reversible PKG inhibitor C2 (kindly donated by Michael Blackman, The Francis Crick Institute) in RPMI complete media to inhibit parasite egress. After drug removal, parasites were resuspended at a concentration of $10^8$ *P. falciparum*-iRBC/mL in EGM-2MV media supplemented with 1% astrocyte and pericyte growth factors (ScienCell), gassed with a mixture of 90% N2, 5% CO2, and 5% O2 and left in the incubator overnight on a shaker (50 rpm) to facilitate parasite egress. The efficiency of parasite egress was assessed by hemocytometer count and blood smear. The concentration was adjusted at $5 \times 10^7$ ruptured *P. falciparum*-iRBC/mL before the suspension was centrifuged at 190 x g for 5 min, aliquoted and flash frozen in liquid nitrogen. The same protocol was applied to uninfected erythrocytes to be used as a negative control medium for the scRNA-seq experiment.

## Sample incubation with *P. falciparum* egress media, Ruxolitinib, *P. falciparum*-iRBC or cytokines

After 6 days in culture, 3D-BBB microvessels were perfused with 150 µL of *P. falciparum*-iRBC egress media under gravity-driven flow and incubated for 24 h at 37 °C. Reservoirs were refilled every 12 h. For Ruxolitinib (Santa Cruz Biotechnology) treatment, 3D-BBB microvessels were co-incubated for 24 h at 37 °C using a 10 µM Ruxolitinib concentration and iRBC-egress media or cell culture media.

For trophozoite and schizont *P. falciparum*-iRBC perfusion, a magnetic cell separation or Percoll gradient was used to purify late-stage parasites at the desired hpi (26–34 hpi for trophozoites or 42–48 hpi for schizonts) at >60% purity. Microvessels were then perfused thrice for 10 min with 150 µL of *P. falciparum*-iRBC at $5 \times 10^7$ iRBC/mL concentration under gravity-driven flow, followed by two consecutive 10-minute washes to remove unbound cells. Devices were then incubated at 37 °C with *P. falciparum*-iRBC for 6 h. As a control, the same concentration of uninfected erythrocytes or uRBC media was used for perfusion and incubation. After incubation, samples were used for single-cell RNA sequencing, permeability assays or immuno-fluorescence staining for confocal imaging with the experimental timeline shown in Figs. 2a and 4a.

Confluent 2D cell monolayers in 8-well slides were incubated with 150 µL of *P. falciparum*-iRBC egress media ($50 \times 10^6$ ruptured iRBC/mL). Cell culture media was used as a control. Alternatively, 2D monolayers were incubated overnight with TNFα (R&D Systems), IL-1β (Peprotech) and interferon γ (IFNγ) (Peprotech), either alone or combined in a cytokine cocktail. All cytokines were used at a concentration of 10 ng/mL. Cell culture media was used as a control.

## Sample preparation and imaging by transmission electron microscopy

Microvessels were pre-fixed for 30 min in 2% PFA / 2.5% glutaraldehyde (GA) in EGM-2MV medium for 30 min and washed thrice for 10 min with EGM-2MV. The collagen hydrogel was carefully removed from the PMMA jig, and the low-shear stress areas of the microvessel network were cut into smaller pieces (about $1 \times 0.5 \times 0.5$ mm) for further processing. The samples were fixed with a secondary fixative solution (2% PFA, 2.5% GA, 0.25 mM $CaCl_2$, 0.5 mM $MgCl$ and 5% sucrose in 0.1 M sodium cacodylate buffer) overnight at 4 ˚C, rinsed twice for 15 min with 0.1 M sodium cacodylate buffer and stained with reduced osmium solution (1% $OsO_4$, 1.5% $K_3FeCN_6$ in 0.065 M Cacodylate buffer) for 2 h at 4 °C. Samples were washed six times for 10 min in distilled $H_2O$ and kept at 4 °C until further processing. Dehydration was performed in steps of 30, 50, 80, and thrice with 100% ethanol in a PELCO Biowave Pro microwave processor (Ted Pella, Inc.) containing a SteadyTemp Pro and a ColdSpot set to 4 °C, each step 40 seconds at 250 W. Samples were infiltrated in serial steps of 25, 50, 75, 90, and twice with 100% EPON 812 hard epoxy resin in acetone, assisted by the microwave (3 min each at 150 W under vacuum) and a final infiltration step in 100% EPON 812 hard epoxy resin overnight at room temperature. Samples were oriented in the embedding mold with the axis of the microvessel lumen at approximately 90° angle to the cutting surface to be able to cut transversal sections of the channels, and then polymerized at 60 °C for 48 h. Microvessel pieces for imaging were randomly selected, and thin sections (70 nm) were retrieved on an ultramicrotome (UC7, Leica Microsystems), collected on formvar-coated slot grids and post-stained in uranyl acetate and lead citrate. Tile montages (12,000x) were acquired on a JEOL JEM 2100 plus at 80 or 120 keV using SerialEM. Montages were processed using IMOD's Blend Montages function and Fiji (ImageJ, v1.54 f). Tight junctions were analyzed as a percentage of length of electron-dense tight junctions over junction length as measured in Fiji (ImageJ, v1.54 f). The percentage of gaps in junctions was counted in Fiji (ImageJ, v1.54 f).

## Microvascular permeability assays

Permeability assays were performed on endothelial-only or 3D-BBB microvessel models after 6 and 7 days in culture. 70 kDa FITC-dextran at a concentration of 100 µg/mL was perfused at a flow rate of 10 µL/min, applied with a withdrawing syringe pump (Harvard Apparatus PHD 2000). Tilescan confocal images were acquired with 5 µm z-step size in 5 different ROIs of the microvessels every 2.5 min over 10 min. For time-lapse experiments, imaging was performed on the same ROI before and after 24 h incubation with *P. falciparum* egress media or 6 h incubation with schizont-stage *P. falciparum*-iRBC.

**Microvascular permeability quantification.** The quantification of apparent microvascular permeability was done using the following formula:

$$Permeability = \frac{1}{\triangle t} \bullet \frac{V_{gel}}{A_v} \bullet \frac{I_{gel1} - I_{gel0}}{I_{v0} - I_{gel0}}$$

The leakage of the fluorescent tracer into the collagen matrix between two time points was calculated as follows, were $\Delta t$ is the time interval between the two frames, $V_{gel}$ is the volume of the collagen matrix, $A_v$ is the lateral vessel surface, $I_{gel1} - I_{gel0}$ is the difference between the fluorescence intensity inside the gel in the two time points, $I_{v0}$ is the fluorescence intensity in vessel at the start of the measurement. An area of determined size (250 µm x 150 µm) containing part of the vessel and the collagen matrix was selected after image analysis in Fiji (ImageJ, v1.54 f). Two ROI were defined, corresponding to the vessel (250 µm x 30 µm) and the adjacent collagen matrix (250 µm x 120 µm) (Supplementary Fig. 1c). For each ROI, fluorescence intensities were obtained at two different time points: t0 (after complete vessel filling with dextran) and t1 (2.5 min later). These values were then used to calculate microvascular permeability (cm/s). Permeability was calculated before incubation with iRBC-egress media or schizont iRBC, and after 24 or 6 h, respectively. A final/baseline permeability ratio was calculated for each microvessel area and used to compare different conditions.

## Statistics

Statistical analysis was performed using GraphPad Prism (version 10.2.0) or R (version 4.2.2). A two-tailed Mann-Whitney U test was used to analyze non-normally distributed samples. The effect size $r$ was calculated as the z-statistic divided by the square root of the sample size $n$. To compare non-normally distributed samples among multiple conditions, the Kruskal-Wallis test with Dunn's multiple comparisons test was used. $P$-values < 0.05 were considered statistically significant. Values are reported as median (IQR), each dot in the graphs represents an ROI or an event from a number (N) of individual wells or independently-fabricated microvessels indicated in the figure legends.

## Sample preparation for single-cell RNA sequencing (scRNA-seq)

The 3D-BBB models were fabricated, perfused and incubated according to the described conditions. 3-4 devices per condition were disassembled, and the main, cell-containing collagen part was dissected. The collagen piece was dissociated for 6–10 min (until complete dissociation) in collagenase diluted in serum-free media (1 mg/mL, Sigma-Aldrich). After stopping the collagenase reaction with complete media, the cells were trypsinized for 8 min to obtain a single-cell solution. The cells were mechanically dissociated by pipetting with wide-bore tips and washed two times in PBS containing 0.1% BSA.

## MULTI-seq sample preparation

The single-cell solutions harvested from the 3D-BBB microvessels were labeled with MULTI-seq barcode oligonucleotides for sample

multiplexing as introduced in ref. 32. Briefly, the cells were resuspended in Cell Prep Buffer (PBS containing 0.1% poly(vinyl alcohol) (PVA) and 1 mM EDTA). A 1:1 mixture of the cholesterol-conjugated Anchor-oligonucleotides (Anchor CMO, synthesized by Integrated DNA Technologies) and Barcode oligonucleotides with a distinct barcode for each sample (final concentration, 0.2 μM) was added, and the cells were incubated on ice for 5 min. Next, the same concentration of Co-Anchor CMO (synthesized by Integrated DNA Technologies) was added and incubated for another 5 min, followed by three washes with PBS containing 1% BSA. The cells in each sample were counted after washing and were combined so that the multiplexed suspension contained the same number of cells from each sample. The combined sample was filtered through a 35 μm cell strainer and counted again before 10x Genomics barcoding. The MULTI-seq barcode sequences used in this study are: *TCCTCGAA* for control uRBC media, *ATGCGATG* for iRBC-egress media, *GCTATGCA* for control RBC, *CGATACTG* for trophozoite stage iRBC, and *TACGCAGT* for schizont stage iRBC.

**10x Genomics barcoding and sequencing.** mRNA transcripts of each cell were barcoded using the Chromium Controller (10x Genomics, firmware version 4.00). The reagent system was Chromium Single Cell 3′ GEM, Library & Gel Bead Kit v3.1 (10x Genomics) and a Chromium Next GEM Chip G Single Cell Kit (10x Genomics). Barcoding and cDNA library construction were performed according to the manufacturer's instructions, and MULTI-seq barcode library preparation was performed as per the MULTI-seq protocol[32]. After the cDNA amplification step, the barcode fraction was collected, amplified, and single-indexed with KAPA HiFi HotStart ReadyMix (Roche) for sequencing. Both finished cDNA and MULTI-seq-barcode libraries were sequenced with NextSeq2000 (Illumina). We read 8 base pairs (bp) for TruSeq Indices, 28 bp for 10x Genomics barcodes and unique molecular identifiers (UMIs), and 52 bp for both fragmented cDNA and MULTI-seq barcodes.

### scRNA-seq data analysis

The scRNA-seq data used in this study are available in the ArrayExpress database under accession code E-MTAB-14463.

**Sequence alignment.** Sequenced reads were aligned to a combined reference genome constructed from the human genome (*GRCh38*) and the *P. falciparum* genome (*hb3*) to generate the feature-barcode matrices with the *CellRanger* pipeline (v. 7.0.1, 10x Genomics). All downstream data analysis was performed with *R version 4.2*[70].

**Demultiplexing of the MULTI-seq sample.** Using the R package *deMULTIplex* (v.1.0.2)[32] we counted the MULTI-seq barcode reads and assigned each cell to being a singlet of a specific condition, a doublet, or a sample-barcode-negative cell. Only singlets were kept for further analysis.

**Quality control (QC).** We performed quality control using the *scuttle* package (v.1.8.4)[71] as follows: Genes that were found in less than 5 cells were excluded. We then identified the QC thresholds based on scatter plots of detected gene counts against the proportion of mitochondrial gene expression in each cell or the *isOutlier* function (nmads=1.5) and only kept the cells above the determined detected gene cutoff (iRBC-egress dataset: 2500, iRBC dataset: 3233/2448) and below the mitochondrial gene cutoff (iRBC-egress dataset: 4%, iRBC dataset: 5%) for further analysis. Subsequently, doublets were excluded using the *scDblFinder* package (v. 1.10.0)[72].

**Data normalization.** We normalized the raw UMI counts of the QC-filtered cells using the deconvolution approach from the R package *scran* (v.1.24.1)[73]. The size factor for the library size correction of each cell was calculated with the *calculateSumFactors* function, and the raw counts were normalized based on the size factor and log2-transformed with the *logNormCounts* function. These values appear as "*logcounts*".

**Identification of highly variable genes (HVG), visualization, and clustering.** After normalization, highly variable genes were chosen with the *modelGeneVar* function in the scran package and genes with a biological variance > 0.1 were chosen as HVGs for further dimension reduction and data integration. Using the chosen HVGs, we performed principal component analysis (PCA) and constructed UMAP plots from the principal components capturing the highest percentage of variance using the *scater* package (v.1.26.1)[71]. The gene expression levels shown in the UMAP plots correspond to the log2-transformed normalized values. Cell population clusters were identified using the *bluster* package (v.1.8.0)[74] and the *Leiden* algorithm[75]. Cell type assignment was performed manually based on the cluster marker genes.

**Data integration.** The dataset of the trophozoite-/ schizont-iRBC perfusion was the only dataset where two 10x reactions were used while both reactions contained all three MULTI-seq-labeled conditions (RBC-/ trophozoite-/ schizont-iRBC perfusion). To integrate the two sequencing runs, the data was scaled according to sequencing depth using the multiBatchNorm function in the R package *batchelor* (v.1.14.1)[76]. HVGs across the datasets were chosen using the *combineVar* function. The *rescaleBatches* function was used for data integration, and the corrected results were used for all visualizations, but not for differential gene expression analysis.

**Differential expression analysis.** Differential expression analysis was performed separately for each cell type utilizing the hurdle (two-part generalized regression) model from the *MAST* package (v.1.22.0)[77]. The Benjamini-Hochberg method was applied to the p-values to account for multiple testing. All significantly differentially expressed genes for all cell types and conditions can be found in Supplementary Data 1. The heatmaps visualizing the log2-transformed fold change (log2FC) values were assembled using the *pheatmap* (v.1.0.12)[78] and the *dendextend* package (v.1.16.0)[79]. The heatmap genes were selected as representatives of the most significantly overrepresented GO-term from the endothelial differentially expressed genes upon iRBC-egress exposure.

**Gene ontology (GO) term over-representation analysis.** GO-term over-representation analysis on significantly up- or downregulated genes was performed separately per each cell type and condition (FDR < 0.05, log2FC > 0.1 or < − 0.1, respectively)) using the *enrichGO* function (Benjamini-Hochberg correction, p-value cutoff 0.05, max. geneset size 1500) from the *clusterprofiler* package (v.4.4.4)[80]. The iRBC-egress dataset included "*Biological Process*" and the iRBC dataset included all GO-term categories. The *pairwise_termsim* and *emapplot* functions from the *GOSemSim* package (v.2.22.0)[81] and the *enrichplot* package (v.1.16.2)[82] were used to visualize the analysis results. GO-term clusters in the enrichment map were manually labeled. The list of all significant GO-terms can be found in Supplementary Data 2.

**Gene signature analysis.** Gene signature scores were calculated using the *tidySingleCellExperiment* package (v.1.6.3)[83]. Log-expression values of all signature genes were summed up, and a signature score was calculated by rescaling the resulting number to a signature score between 0 and 1 for every single cell:

$$MHC\ signature\ score = rescale(sum(log\ gene\ expression\ values),\ to = c(0,1))$$

The genes included in the MHC signature scores were obtained from the KEGG pathway[84] "*hsa04612-Antigen processing and presentation*". These include:

MHC1: *PSME3, PDIA3, HLA_A, HLA_B, HLA_C, HLA_E, HLA_F, HLA_G, HSPA1A, HSPA1B, HSPA1L, HSPA2, HSPA4, HSPA5, HSPA6, HSPA8, HSP90AA1, HSP90AB1, B2M, PSME1, PSME2, TAP1, TAP2, TAPBP, CALR, CANX*;

MHC2: *IFI30, CREB1, CTSB, CTSL, CTSS, HLA_DMA, HLA_DMB, HLA_DOA, HLA_DOB, HLA_DPA1, HLA_DPB1, HLA_DQA1, HLA_DQB1, HLA_DRA, HLA_DRB1, HLA_DRB5, CIITA, NFYA, NFYB, NFYC, LGMN, RFX5, RFXAP, RFXANK, CD74*

The *iRBC-egress signature score* was obtained, as described above, by separately calculating a signature score of the 50 most upregulated genes and the 50 most downregulated genes (adjusted p-value < 0.05 & lowest log2FC) in endothelial cells, upon exposure to iRBC-egress media (genes marked in Supplementary Data 1). The total iRBC-egress signature score was calculated by subtracting the score of the down-regulated genes from the upregulated genes:

$$iRBC\ egress\ signature\ score$$
$$= rescale(sum(log\ gene\ expression\ 50\ most\ upregulated\ genes),$$
$$to = c(0,1)) - rescale(sum(log\ gene\ expression\ 50\ most\ downregulated\ genes),$$
$$to = c(0,1))$$

The threshold (sum of log2-transformed normalized *P. falciparum* gene counts > 10) for defining the *P. falciparum*-RNA-high endothelial cell population among the schizont-iRBC-exposed cells (Fig. 7) was determined based on the background *P. falciparum*-RNA level in the RBC control.

**Signaling pathway activity analysis.** Signaling pathway activities were calculated using the *PROGENy* package (v.1.18.0)[37] (using the top 500 genes for generating the model matrix according to significance). The obtained progeny scores were summarized per cell type and condition and visualized using the *pheatmap* package.

**Ligand-receptor interaction analysis.** Ligand-receptor interaction analysis was performed using the *CellChat* package (v.1.6.1)[77]. *Cellchat* was run on cells from either of the two experimental conditions separately before the *Cellchat* objects were merged and the interactions compared. Up- and down-regulated ligand-receptor signaling pairs were identified from the differential expression analysis using the *identifyOverExpressedGenes* function (thresh.fc = 0.1, thresh.*p* = 0.05). Chord diagrams were constructed showing selected differential receptor-ligand interactions involved in known BBB-specific interactions (Figs. 2, 4) or all identified differential receptor-ligand interactions (Supplementary Fig. 3).

**Reporting summary**
Further information on research design is available in the Nature Portfolio Reporting Summary linked to this article.

## Data availability
The scRNA-seq data used in this study are available in the ArrayExpress database under accession code E-MTAB-14463. Source data are provided in this paper.

## Code availability
The code generated during this study is available at https://github.com/Alina-Ba/scRNAseq_iRBC.

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

## Acknowledgements

We want to thank all members of the Bernabeu lab for supportive discussions and critical feedback. Furthermore, we would like to thank Kristina Haase for her suggestions on the project and Violeta Beltran-Sastre for her support in the generation of fluorescently labeled cells. We would like to thank Sergi Beneyto Calabuig, Lars Velten, Wolfgang Huber, Dewi Moonen, and Dominik Lindenhofer for their input on the scRNA-seq experiments and data analysis. We are grateful to the Genomics Core (GeneCore) facility at EMBL, Genomics Core facility at the Universitat Pompeu Fabra (UPF), the Genome Biology Computational Support (GBCS) (Charles Giradot) at European Molecular Biology Laboratory (EMBL) for their sequencing support and to EMBL IT Service for the service of high-performance computing resources. We thank the Electron Microscopy Core Facility (EMCF) at EMBL for their support. The great majority of this work was funded by the European Research Council (ERC) under the European Union's Horizon 2020 research and innovation program (Grant agreement no. 948088), the EMBL core program funding and contributions from the EMBL Infection Biology Transversal Theme. F.K. is funded through the Marie Skłodowska-Curie grant agreement (101068552). H.F. is supported by a fellowship from the EMBL Interdisciplinary (EIPOD4) program under Marie Skłodowska-Curie Actions COFUND (847543).

## Author contributions

L.P., A.B., and M.B. conceived the work. L.P., A.B., H.F., F.N., F.K., R.K.M.L., and M.B. designed experiments. L.P., A.B, H.F., F.N., F.K., and R.K.M.L. performed experiments with the assistance of T.R., B.L., and S.S. L.P., A.B., H.F., F.K., and T.R. analyzed the experimental data under the guidance of Y.S. and M.B. A.B and F.N. performed the scRNA-seq analysis with input from D.S. and J.A.H on experimental planning and analysis, under the guidance of L.M.S., J.S., and M.B. L.P., A.B., and M.B. wrote the original draft of the manuscript. All authors contributed to manuscript writing, revision, editing and suggestions. M.B. contributed to project supervision, administration and funding acquisition.

## Funding

## Competing interests

The authors declare no competing interests.
