## [Transparent Peer Review file · Nature Communications]

***Plasmodium falciparum* egress disrupts endothelial junctions and activates JAK-STAT signaling in a microvascular 3D blood-brain barrier model**

Corresponding Author: Dr Maria Bernabeu

Version 0:

Reviewer comments:

Reviewer #1

(Remarks to the Author)

The manuscript by Piatti et al presents a bioengineered microvascular model to study the blood brain barrier (BBB) in the context of cerebral malaria. The model is significantly more complex than previous systems and shows major physiological hallmarks of the BBB. The authors also demonstrate how infected red blood cells, and their products (egress medium) alter endothelial barrier function and induce a differential cellular response in the three cell types composing the bioengineered BBB.

Altogether the data presented in this manuscript represent a significant step forward in the development of reductionist models recapitulating one of the major pathological features of human malaria, i.e. cerebral malaria.

The MS is well written, and experimental data are solid with appropriate controls. I have only minor comments.

- Figure 1 and thereafter: the use of egress medium is referenced from a preprint from the same authors (Long et al). There are no other details provided about this essential tool used here. Do the authors assume the properties are driven by parasite products such as Hz, histones, etc? Could they add something in the discussion along these lines?
- Figure 2h: % gaps in junctions. Is this difference significant or not? Please clarify.
- Figure 3a. Is the EC graph the same as in 2e? Please clarify.
- Figure 3b. It would be helpful if cell types are colour coded for clarity (red, blue green?)
- Figure 3h,i. Pericytes and astrocytes do not show increased STAT1 nuclear localization. This seems to contradict the scRNAseq data showing an upregulated JAK-STAT pathway in these cell types. Please clarify.
- Figure 4b. What system are the Pf annotations such as PFHG_02607? These are not PlasmoDB accession numbers.
- Figure 4 and elsewhere. Jambou et al described a trophocytosis-like process during interaction of iRBCs and ECs. This paper is not cited and the process not observed here. A few phrases in the discussion would be helpful to discuss/explain this discrepancy.
- Figure 5h. ECs apparently containing iRBCs based on scRNAseq read counts show an increase in the egress signature score. Can the presence of iRBCs in ECs be confirmed independently by IFA and/or FISH?
- Line 360: agreeance should be agreement

Reviewer #2

(Remarks to the Author)

Pathogenic mechanisms of *Plasmodium falciparum* egress unveiled by a microvascular 3D blood-brain barrier model by Livia Piatti et al.,

Novelty: Bioengineered 3D microvascular model of the BBB incorporating astrocytes, ECs and pericytes to model *P. falciparum* infection in vitro and study the consequences on BBB in the context of cerebral malaria. In this study, the authors developed a bioengineered microvascular 3D-BBB model that incorporates primary human brain microvascular endothelial cells, pericytes and astrocytes. The addition of these cell types increased the vascular barrier function of the model, improving upon other bioengineered models previously used to study *P. falciparum* pathogenesis.

Major Findings: Merozoite-stage red blood cell egress products downregulate express of junction proteins and increase permeability; induce inflammatory and antigen presentation pathways; however, binding of intact red blood cells infected with trophozoite or schizont stages (pre-rupture of the RBC membrane) have relatively minor effects on barrier integrity and transcriptional responses of the vascular cells in this model.

Overall, this is a very promising model for understanding host parasite interactions at the interface between blood and the brain where the most critical steps in disease pathogenesis occur. However, there are several over-arching issues that need to be addressed. First, it is unclear whether it is parasite products (metabolites, proteins, lipids, nucleic acids, etc) that are responsible for the disruption of barrier integrity and the activation of vascular cell transcriptional responses, or is it simply the products of red blood cell destruction? To address this, it may be important to include a comparator of uninfected red blood cell lysate. This is important because hemolysis (both direct and by-stander hemolysis) has been proposed as a major mechanism of disease in malaria and this model could address that and provide more clarity around the relative contributions of RBC destruction versus receptor ligand interactions on endothelial integrity and transcriptional responses. Second, as detailed below, many of the experiments presented here require additional validation beyond the transcriptional data provided. This may include validating protein targets and providing additional images of the cell models stained with relevant markers.

Major Comments:

Most of the data are provided at the mRNA level (scRNAseq analysis), it would be interesting to validate protein expression levels via IF studies of tight junction proteins: ZO-1, claudin5 and occludin are needed. Although mRNA levels are differentially expressed in ctrl vs infected conditions, it would be interesting to see if there are any defects in junctional proteins organization.

In Fig 1c, the authors provide immunofluorescence images of the 3D BBB, were these images acquired at day 7 of the protocol? The timeline appears in Fig.2a. Perhaps provide a schematic of the timeline in Fig.1 as well so the reader knows at what time point the images were performed. Similar images as in fig.1C are missing when the 3D BBB culture is exposed to the egress media (staining for GFP-HA, mCH-HBVP, VeCad, α -sma). Fig. 2j, provides images only for DAPI and VeCad, astrocytes and pericytes staining are lacking. Also, it would be necessary to address EC, pericytes and astrocytes survival/apoptosis in each condition (3D BBB+/- egress media).

In Figure 3, the authors observed an increase in VEGF levels in ECs and pericytes, it would be interesting to validate with Western Blot analysis as a functional read out for VEGF signaling one of the VEGF signaling targets, for example, eNOS levels as it is known that CM is associated with eNOS dysfunction. WB analysis of VEGFR2 downstream effectors namely ERK and AKT are also required to validate scRNAseq findings on EC monolayer +/- egress media. Additionally, the increase in VEGF levels in pericytes seems less significant than in ECs.

In Fig 3D, the authors mentioned the activation of autophagy from observation of TEM images. This statement is descriptive, a functional assay for autophagy is required to validate this observation.

General comment Fig 3: As a therapeutic strategy and to validate the role of JAK/STAT pathway in their 3D BBB +/- egress media model, it would be interesting to test the effect of the drug Ruxolitinib in their model to see if BBB integrity and function in CM is improved (permeability, junctional proteins, inflammation).

In extended Fig 3: the downregulation of Notch signaling pathway downstream effectors such as DLL4, JAG1/2 shown in the heatmap is not very convincing, same for ANGPT1/2

In Fig4: when the authors use a different life stage of the parasite (e.g., trophozoite) they observe downregulation of genes regulating blood vessel development such as APLN, ROBO4, etc; However, in the egress media model, they observed different pathways namely NOTCH, JAK/STAT, VEGF, is there any regulation of downstream effectors of these pathways with the trophozoite/schizonts model?

CD8+ T cells have been identified as key players in the disruption of the endothelial barrier and targeting these cells specifically is currently in clinical trials for cerebral malaria in African children. How do you integrate or reconcile the current findings with this well-established mechanism? Do you envision them as independent events, or is the uptake and presentation of parasite products via MHC Class I perhaps the target of TCR-mediated destruction of endothelial cells by CD8+ T cells? This concept needs to be discussed.

Minor comments:

In abstract provide abbreviation form of blood brain barrier (BBB) when first used in line 24

Describe in full EPCR and ICAM1 (line 47, Main)

Fig 2K: add in result section that permeability was assessed by FITC-Dextran 70KDa (although it is written in the legends, it will be easier for the reader to know what assay was used) Line 147

In Figure 3e, please provide the Mann-Whitney p values.

In Figure 3j, the legend indicates magenta for ICAM-1, but there is no magenta visible. Instead, there is a light blue stain that appears to mark nuclei – is this DAPI? Also, the prominent feature appears to be loss of nuclei. Also, the magnification

appeared to change. Please examine this figure closely and correct any issues.

While the egress media reflects a calculated parasite density, what do the authors speculate will be the concentration of egress products in the blood circulation, and might there be local variation in this concentration? In other words, how closely do these experimental conditions reflect the levels that endothelial cells are exposed to in vivo during a malaria infection?

In some experiments PECAM1 is used for normalization. It is important that PECAM1 expression itself is not affected by the egress media.

Reviewer #3

(Remarks to the Author)

The manuscript presents a 3D blood-brain barrier model to understand how *P. falciparum* parasites affect brain vasculature using primary human cells (HBMECs, astrocytes, and pericytes) with basic validation through immunofluorescent staining of cell-specific markers and tracking via lentiviral labeling. The key findings revealed that when exposed to *P. falciparum* egress products, the BBB model showed downregulation of critical endothelial junction and vascular development genes, leading to increased microvasculature permeability. The study also found that while short-term exposure to infected red blood cells caused more localized changes, both conditions compromised BBB integrity. This timely and necessary study offers valuable insights into the mechanisms of BBB disruption in cerebral malaria. Despite the availability of effective drugs, the lack of adjunctive therapy to prevent severe complications of cerebral malaria remains a critical limitation in recovery for survivors of this deadly disease.

The innovative bioengineered 3D model demonstrates functional barrier properties and cell-type-specific responses. However, the main concern is that the authors do not mention any limitations of the primary cells used in the study. Overall, the manuscript is well-written and logically organized. The authors have done a good job characterizing iRBCs and addressing barrier permeability questions, and the visualization and quantification of cell-specific markers are comprehensive.

Specific points to consider:

As with any study utilizing HBMECs, additional information about the primary cells used would be helpful. Were these pooled from multiple donors? Was the same batch of cells used for all experiments? This is unlikely, considering the amount of work presented. In that case, was some validation done between lots of cells used? These additional details would provide crucial context for the reproducibility and reliability of the findings.

The study significantly advances BBB modeling for cerebral malaria with 3D microvascular fabrication. The authors do address the potential benefits of iPSC-derived cells for better standardization in their discussion. The use of such a model is largely absent in cerebral malaria. They also mention the report by Lu et al. (2021), which challenges the iPSC-BBB model proposed by Lippmann et al. (Nat. Biotech, 2012). However, the iPSC model has since been further refined, evaluated, and widely applied over the last decade. If the authors want to mention the Lu et al. study as a limitation of iPSCs, they should note that the model proposed by the group utilized the overexpression of ETS transcription factor transgenes from lentiviral vectors and lacked several critical barrier phenotypes. iPSCs have other limitations that the authors can address.

Minor points:

Additional details on optimizing cell ratios for constructing the 3D model, including more information on the QC metrics for the 3D model, would be beneficial. This information, combined with previously mentioned data on the reproducibility of primary cell donor findings, would address any validity concerns about the study.

Reviewer #4

(Remarks to the Author)

Version 1:

Reviewer comments:

Reviewer #1

(Remarks to the Author)

The authors have addressed all of my comments. I have no further requests or comments.

Reviewer #2

(Remarks to the Author)

We thank the authors for the careful consideration of all the reviewers' suggestions and questions. Several of the key points we raised were directly implemented in the revision. Notably, the authors incorporated the suggestion to test the effect of the JAK/STAT inhibitor (ruxolitinib) in their 3D system with egress media. Their new Figure 6 provides compelling data showing that ruxolitinib helps restore tight junction integrity and improves barrier permeability—an important advancement in understanding how this pathway could be targeted therapeutically in cerebral malaria (CM). We endorse the authors' revised title that better reflect the central role of JAK/STAT signaling.

Another critical point raised concerned the validation of their ultrastructural findings. We recognize the new LAMP1 staining experiments to confirm lysosomal uptake of parasitic material by endothelial cells, strengthening interpretation of the TEM data. We appreciate the additional findings on astrocyte behavior in the presence of egress media, which suggests a mechanical stabilizing role at sites of endothelial disruption.

Overall, the authors' additional efforts have significantly strengthened this work and its translational potential. The revised paper now more clearly highlights the therapeutic relevance of JAK/STAT inhibition in CM, particularly the potential to repurpose ruxolitinib in combination with antimalarials to restore blood-brain barrier (BBB) function. While further *in vivo* studies are still necessary to assess neurobehavioral outcomes, the 3D model remains a robust and promising tool for therapeutic testing.

Nafiisha Genet, PhD (UVA)
Hans Ackerman, MD DPhil (NIAID)

Reviewer #3

(Remarks to the Author)

The authors have adequately addressed my concerns about the previous iteration of the manuscript. I appreciate the authors' detailed response to each review question and the inclusion of useful additional data.

A couple of important new questions arise based on new data included in the revised manuscript.

1. The earlier iteration discussed differential mRNA expression only. In the revised manuscript, the newly added Figures 6a-c raise questions about TJ protein expression and localization by IF.

The authors were able to stain for ZO-1, but not for Claudin-5 or occludin, and offered a few explanations in the rebuttal, but have not addressed a critical missing data point at all in the manuscript beyond mentioning increased mRNA expression.

A known limitation of primary HBMECs is the reduced expression of claudin-5 compared to microvessels, which is a key driver of the low junctional tightness of these primary cells. Gericke et al. (2020) is more recent among the studies attempting to address the limitation of low Claudin-5 expression in HBMECs.

Figure 6 b shows that Ruxolitinib increases STAT1 expression and rescues ZO-1, but without Claudin-5, the specific ability to restore BBB integrity is questionable.

ZO-1 is a scaffolding protein, Claudin-5 is the primary transmembrane protein of the BBB, and occludin modulates barrier stability.

While the focused title highlighting the activated JAK-STAT pathway is reasonable, the study has not shown sufficient evidence of BBB junctional disruption.

The claim that Ruxolitinib preserves BBB integrity would be strengthened by post-treatment assessment of claudin-5 localization and expression.

2. Relatedly, the absence of EPCR/ICAM-1 binding validation is a major omission. The earlier draft included ICAM-1 expression, which I misunderstood to be on the surface of HBMECs. A response to a reviewer's query clarified that it was about astrocytes.

3. In response to Reviewer 2's comment about egress media as a correlate for parasite density. The authors suggest 5×10^4 ruptured iRBCs/uL correspond to about equal numbers of circulating parasitemia values of 5×10^4 parasites/uL or 1%, which are frequently found in patients. This is mathematically consistent if we assume 1 iRBC = 1 parasite. However, late-stage schizonts contain anywhere from 16 to 24 merozoites.

The authors' referenced manuscript (Long et al., 2024, bioRxiv preprint) details the methodology for blocking iRBC egress and the use of late-stage schizonts. There is no mention of testing higher or lower dilutions of iRBCs used, so the concentration used is only justified by the math above, which does not account for merozoites.

Even accounting for circulating parasitemia in clinical infections being an underestimate of the actual parasite density (due to sequestered parasites), using mature schizonts, the egress media likely delivered a significantly higher parasitemia than would be seen in the most severe clinical cases.

4. With the additional Ruxolitinib experiments, the authors can show some other measures of baseline vs. iRBC exposed vs.

drug

Reviewer #4

(Remarks to the Author)

REVIEWER COMMENTS

Reviewer #1 (Remarks to the Author):

The manuscript by Piatti et al presents a bioengineered microvascular model to study the blood brain barrier (BBB) in the context of cerebral malaria. The model is significantly more complex than previous systems and shows major physiological hallmarks of the BBB. The authors also demonstrate how infected red blood cells, and their products (egress medium) alter endothelial barrier function and induce a differential cellular response in the three cell types composing the bioengineered BBB.

Altogether the data presented in this manuscript represent a significant step forward in the development of reductionist models recapitulating one of the major pathological features of human malaria, i.e. cerebral malaria.

The MS is well written, and experimental data are solid with appropriate controls. I have only minor comments.

We thank the reviewer for the supportive comments on our manuscript. We appreciate the recognition of the relevance of our model for studying cerebral malaria and the positive evaluation of the experimental design and data.

We have addressed the reviewer's suggestions and implemented the changes in the manuscript as detailed below.

Figure 1 and thereafter: the use of egress medium is referenced from a preprint from the same authors (Long et al). There are no other details provided about this essential tool used here. Do the authors assume the properties are driven by parasite products such as Hz, histones, etc? Could they add something in the discussion along these lines?

Thank you for your suggestion. We have expanded our hypothesis on the disruptive elements of egress media in the discussion, as well as a schematic representation of the egress media preparation protocol (Extended Data Fig. 2a):

Lines 384-386

“Our study confirms that the egress of sequestered *P. falciparum* parasites in close proximity to endothelial cells is a key pathogenic event, accompanied by the release of agents such as heme¹⁷, hemozoin¹⁸, parasite histones^{19,20}, PfHRP2²¹, and GPI anchors²³, which have been previously described to activate endothelial cells and compromise barrier integrity.

Figure 2h: % gaps in junctions. Is this difference significant or not? Please clarify.

We have analyzed more sections from TEM imaging and we have observed a significant difference in the percentage of gaps at junctions. The new graph can be found in Fig 2i (bottom graph).

Figure 3a. Is the EC graph the same as in 2e? Please clarify.

Yes, in Fig 2e we focus on downregulated genes while in Fig 3a we focus on upregulated genes. We clarified in the figure legend:

Line 1059-1063

“a, Volcano plots of differentially expressed genes in endothelial cells (same analysis as in 3e), pericytes, and astrocytes upon 24-hour incubation with iRBC-egress media plotting the log₂-FC against the statistical significance (-log₁₀ of the FDR). Significantly up- or downregulated genes (FDR<0.05, log₂FC>0.1 or log₂FC<-0.1, respectively) are marked in red or blue and selected upregulated genes are labeled.”

Figure 3b. It would helpful if cell types are colour coded for clarity (red, blue green?)

This graph only reflects GO-term over-representation analysis on significantly upregulated genes in endothelial cells. We have clarified this in the graph and corresponding figure legend, as follows:

Line 1063-1067:

b, GO-term over-representation analysis on significantly upregulated genes (FDR<0.05, log₂FC>0.1) in endothelial cells. Each network node represents one of the most significant GO-terms (adjusted p-value < 0.0001) and edges connect GO-terms with more than 20% gene overlap. GO-term clusters were manually summarized in one label term.

Figure 3h,i. Pericytes and astrocytes do not show increased STAT1 nuclear localization. This seems to contradict the scRNAseq data showing an upregulated JAK-STAT pathway in these cell types. Please clarify.

Indeed, we do not observe increased nuclear localization of STAT1 in astrocyte and pericyte monolayers upon exposure to iRBC-egress media. However, the cell growth in a 3D environment with three different cell types and physiological mechanical cues may account for the different response we observe. We hypothesize that the cross-talk between the three cell types might be crucial to account for differences in the 2D and 3D experimental setup.

Notably, in our 3D-BBB model we do observe an overall increased STAT1 fluorescent signal in cells embedded in collagen (i.e. astrocytes and pericytes) after incubation with iRBC-egress media, whereas this signal is barely detected under control conditions. Unfortunately, due to limitations in our immunofluorescence protocol and imaging setup, including limitations with objective working distance using higher magnification objectives and difficulties to getting antibodies through the collagen in an immunofluorescence experimental microfluidic setup, we are unable to achieve sufficient resolution to assess STAT1 nuclear translocation in our 3D microvessels.

Figure 4b. What system are the Pf annotations such as PFHG_02607? These are not PlasmoDB accession numbers.

The annotations were derived from Ensembl Protists. We have added the PlasmoDB identifiers in the main text, in Extended Data Fig. 5a, and the respective figure legend.

Line 251 and 252:

“The UMAP confirmed the correct synchronization and development of *P. falciparum*-iRBC stages, as visualized by a trophozoite cluster positive for the *P. falciparum* mid-stage transcript PfHB3_100020300 and a continuous, arch-shaped schizont cluster positive for late-stage marker PfHB3_090035000 (Fig 4b and Extended data Fig 5a).”

Figure 4 and elsewhere. Jambou et al described a trogocytosis-like process during interaction of iRBCs and ECs. This paper is not cited and the process not observed here. A few phrases in the discussion would be helpful to discuss/explain this discrepancy.

Thank you for your suggestion. In this study we have and mostly focused and study in depth the blood stages related to egress of the parasite, as we found that this stage is the most functionally barrier disruptive and the one causing the most transcriptional effects on the BBB model with the methodology we used in the manuscript. Nevertheless, we recognize the importance and seminal contribution of Jambou et al. Indeed, the formation of a transmigratory-like cup upon binding of *P. falciparum*-iRBC is currently being investigated by another member of our group in a separate project using volume electron microscopy approaches, as this technique is better suited to observed subcellular and ultrastructural changes on the endothelial cell morphological changes that occur early after cytoadhesion.

However, we have now included a reference to the results observed in the paper by Jambou et al in two sections of the discussion:

Lines 377-380

“We cannot rule out the possibility of a *P. falciparum* binding-induced transcriptional shift at later time points, and the occurrence of non-transcriptional cellular processes not evaluated in our study, including morphological or mechanical modifications on endothelial cells, such as trans migratory-like cup structures or trogocytosis, previously identified in *P. falciparum* in vitro studies.”

Line 416-419

“Our findings and others^{44,52,53} suggest a potential mechanism for leukocyte recruitment not only intravascularly by endothelial cells³⁰, but also at the brain perivascular space, aligning with recent observations of vascular and perivascular accumulation of CD8+ T-cells in the brain microvasculature in post-mortem samples of CM patients^{54,55}”

Figure 5h. ECs apparently containing iRBCs based on scRNAseq read counts show an increase in the egress signature score. Can the presence of iRBCs in ECs be confirmed independently by IFA and/or FISH?

The scRNAseq counts of *Pf* mRNA are an indication of egress parasite material (including RNA) uptake by endothelial cells. As mentioned in the reply above, we do not show any engulfment of intact iRBCs. Nevertheless, in Fig 3d and Extended Data Fig.4a we do show by TEM that parasite and red blood cell products present in egress media are taken up by endothelial cells after exposure to iRBC-egress media. Furthermore, to address this concern and a related one expressed by Reviewer 2, in the revised manuscript we now show in Fig. 3e that the lysosome marker LAMP1 colocalizes with the parasite pigment hemozoin and nucleic acids, suggesting that the parasite material uptake occurs via the endothelial lysosome compartment.

Line 360: agreeance should be agreement

Thank you for spotting this mistake. We have replaced ‘agreeance’ with ‘agreement’ in the text (Line 398).

Reviewer #2 (Remarks to the Author):

Pathogenic mechanisms of *Plasmodium falciparum* egress unveiled by a microvascular 3D blood-brain barrier model by Livia Piatti et al.,

Novelty: Bioengineered 3D microvascular model of the BBB incorporating astrocytes, ECs and pericytes to model *P. falciparum* infection in vitro and study the consequences on BBB in the context of cerebral malaria. In this study, the authors developed a bioengineered microvascular 3D-BBB model that incorporates primary human brain microvascular endothelial cells, pericytes and astrocytes. The addition of these cell types increased the vascular barrier function of the model, improving upon other bioengineered models previously used to study *P. falciparum* pathogenesis.

Major Findings: Merozoite-stage red blood cell egress products downregulate express of junction proteins and increase permeability; induce inflammatory and antigen presentation pathways; however, binding of intact red blood cells infected with trophozoite or schizont stages (pre-rupture of the RBC membrane) have relatively minor effects on barrier integrity and transcriptional responses of the vascular cells in this model.

Overall, this is a very promising model for understanding host parasite interactions at the interface between blood and the brain where the most critical steps in disease pathogenesis occur. However, there are several over-arching issues that need to be addressed. First, it is unclear whether it is parasite products (metabolites, proteins, lipids, nucleic acids, etc) that are responsible for the disruption of barrier integrity and the activation of vascular cell transcriptional responses, or is it simply the products of red blood cell destruction? To address this, it may be important to include a comparator of uninfected red blood cell lysate. This is important because hemolysis (both direct and by-stander hemolysis) has been proposed as a major mechanism of disease in malaria and this model could address that and provide more clarity around the relative contributions of RBC destruction versus receptor ligand interactions on endothelial integrity and transcriptional responses. Second, as detailed below, many of the experiments presented here require additional validation beyond the transcriptional data provided. This may include validating protein targets and providing additional images of the cell models stained with relevant markers.

We thank the reviewer for the careful and detailed assessment of our manuscript. We appreciate the recognition of the value and potential of our 3D microvascular BBB model for investigating *P. falciparum* pathogenesis, as well as the suggestions to improve the clarity and rigor of the study. We believe that the additional experiments and clarifications have strengthened the overall conclusions and improved the robustness of the study.

We would like to thank the reviewer for the suggestion of considering the role of hemolysis in the pathogenesis in cerebral malaria. We have added heme as a potential barrier disruptive driver in the introduction (Line 54). Regarding the main issue raised by this reviewer, we would like to clarify that our iRBC-egress media was generated using an optimized protocol specifically designed to mimic the natural release of *P. falciparum* egress products. This approach uses highly synchronized late stage schizonts, a stage of the parasite with lower hemoglobin content due to the parasite metabolism. Our protocol also minimizes mechanical or non-specific red blood cell lysis, compared to other freeze and thawing methods to generate lysates. We believe that our method reduces the release of free hemoglobin or other byproducts of hemolysis. Nevertheless, we agree that there might be residual heme as a consequence of hemolysis derived from blood storage. We have added a new

supplemental figure that better illustrates the protocol for the generation of egress media (Extended Data Fig 2a).

Regarding the controls used in the previous submission, we would like to apologise, as we mentioned that the uRBC control media was only used in the scRNA-seq experiments. However, we should have mentioned that it was also used as a control for TEM and immunofluorescent imaging. The only experiment that did not use uRBC was the permeability assay. Therefore, to determine whether uRBC control causes changes in barrier disruption a 2D impedance-based assay (xCELLigence), providing a continuous readout during a 24h period, and at 0h and 24h in the 3D-BBB model.

We could not see any significant differences in the microvascular permeability ratio of 3D-BBB devices incubated with uRBC control media compared to the media control condition after 24h incubation. The 3D-BBB model permeability results are now shown in Extended Data Fig. 2d, along with another control requested by Reviewer 3 to show consistency across primary endothelial cell donors. We have included it the manuscript as follows:

Lines 159-161

“Notably, no significant differences in microvascular permeability ratio were observed between the two HBMEC donors used in this study (Extended data Fig. 2c), nor in 3D-BBB microvessels incubated with uRBC media control (Extended Data Fig. 2d).”

Please, find below a point-by-point response to the further concerns raised by this reviewer.

Major Comments:

Most of the data are provided at the mRNA level (scRNAseq analysis), it would be interesting to validate protein expression levels via IF studies of tight junction proteins: ZO-1, claudin5 and occludin are needed. Although mRNA levels are differentially expressed in ctrl vs infected conditions, it would be interesting to see if there are any defects in junctional proteins organization.

Thank you for your suggestion. We have performed an IF staining of ZO-1, in which we compared its organization upon exposure to iRBC-egress media or control media. In line with our scRNA-seq results, we found an altered ZO-1 organization, resulting in inter-

endothelial gaps and reduced and fragmented junctional localization. This result can be found in a new panel in Figure 6b. Unfortunately, we could not show staining with tight junction proteins claudin5 and occludin because several tested antibodies showed poor-quality staining for these markers. This is quite likely a consequence of a lower expression of these markers by primary endothelial cells and the difficulties of performing IF staining under flow. Furthermore, methanol fixation, which is widely used in the vascular biology field as it better preserves the structure of the epitopes, is not compatible with our 3D microvascular model.

In Fig 1c, the authors provide immunofluorescence images of the 3D BBB, were these images acquired at day 7 of the protocol? The timeline appears in Fig.2a. Perhaps provide a schematic of the timeline in Fig.1 as well so the reader knows at what time point the images were performed.

Thank you for the suggestion to include a timeline in Fig.1a, which we have now added for more clarity.

Similar images as in fig.1C are missing when the 3D BBB culture is exposed to the egress media (staining for GFP-HA, mCH-HBVP, VeCad, a-sma). Fig. 2j, provides images only for DAPI and VeCad, astrocytes and pericytes staining are lacking. Also, it would be necessary to address EC, pericytes and astrocytes survival/apoptosis in each condition (3D BBB+/- egress media).

We have added more images showing GFAP and aSMA staining after exposure to egress media, now in Figure 3k. We would like to point out that the effect of egress products in pericytes has extensively been covered in another pre-print of the lab. There, we have shown that iRBC-egress media causes low morphological changes in pericytes (analyzed by volume EM). Nevertheless, egress products completely halt the secretion of angiopoietin-1 by pericytes, suggesting a key role of these cells in the disruption of the Ang-Tie2 axis in cerebral/severe malaria.

To address the reviewer's suggestion, we have further investigated the effect of iRBC-egress media on astrocytes in our 3D-BBB. While an increase in astrocyte reactivity was not found (lack of GFAP overexpression), we have observed that astrocytes seem to mechanically hold endothelial cells in place and bridge regions with major a degree of endothelial disruption. We now show some representative images in Fig. 3k:

Lines 233-236

“Despite the lack of astrocyte reactivity after exposure to *P. falciparum*-egress media, astrocytes were often found to extend their processes towards regions of endothelial disruption in our 3D-BBB model (Fig. 3k).”

We attempted to assess apoptosis of BBB cell types using IF staining for cleaved caspase-3. However, the results were inconclusive. We suspect this may be due to elevated background fluorescence of this antibody in our 3D-BBB model and in the presence of egress media, which can obscure detection of low-abundance or highly specific targets such as cleaved caspase-3.

In Figure 3, the authors observed an increase in VEGF levels in ECs and pericytes, it would be interesting to validate with Western Blot analysis as a functional read out for VEGF signaling one of the VEGF signaling targets, for example, eNOS levels as it is known that CM is associated with eNOS dysfunction. WB analysis of VEGFR2 downstream effectors namely ERK and AKT are also required to validate scRNAseq findings on EC monolayer +/- egress media. Additionally, the increase in VEGF levels in pericytes seems less significant than in ECs.

We thank the reviewer for this thoughtful comment. While we did observe a significant increase in VEGF expression in both endothelial cells (logFC = 0.58) and, to a lesser extent, in pericytes (logFC = 0.14), we did not detect corresponding upregulation of downstream effectors such as ERK, AKT, or eNOS in our scRNA-seq data. We attempted to validate these findings by Western blot analysis. However, the results were inconclusive and no differences were found between egress media and control.

Given that VEGF signaling is not a primary focus of our study and the data on its downstream pathways are not robust, we agree with the reviewer that it would be more appropriate to remove emphasis on this pathway from the manuscript. Accordingly, we have removed the sentence discussing VEGF expression in the revised version.

In Fig 3D, the authors mentioned the activation of autophagy from observation of TEM images. This statement is descriptive, a functional assay for autophagy is required to validate this observation.

Thank you for pointing this out. We initially use the term autophagy because the multiple TEM images that we have analyzed show human organelles (e.g. mitochondria with cristae) within these vesicles. To validate this quantitatively, we performed an additional staining with the autophagy marker LC3 and did not observe a positive staining on endothelial cells upon incubation with iRBC-egress media. Nevertheless, we have explored other endocytic pathways and performed an IF staining for the lysosomal marker LAMP1, now shown in Fig. 4e. We observed colocalization of the LAMP1 signal with the parasite DAPI signal, confirming the uptake of parasite material into the lysosomal compartment of endothelial cells.

e

We have rephrased the text as follows:

Lines 196-201:

“Endothelial cells in the 3D-BBB microvessels incubated with iRBC-egress media showed signs of activation, including the formation of membrane protrusions, large vacuoles containing iRBC membranes or hemozoin (Fig. 3d and Extended Data Figure 4a), indicating the activation of parasite uptake. Immunofluorescent staining shows colocalization of parasite nucleic acids with lysosome-associated membrane protein 1 (LAMP1) (Fig. 3e), suggesting the delivery of parasite material to the lysosomal compartment of endothelial cells.”

General comment Fig 3: As a therapeutic strategy and to validate the role of JAK/STAT pathway in their 3D BBB +/- egress media model, it would be interesting to test the effect of the drug Ruxolitinib in their model to see if BBB integrity and function in CM is improved (permeability, junctional proteins, inflammation).

We would like to thank the reviewer for the suggestion to further investigating the role of the JAK/STAT pathway in cerebral malaria through its inhibition. We have added a new figure that shows that co-incubation Ruxolitinib protects the BBB disruption mediated by iRBC egress products. In this figure we show that ruxolitinib prevents the increase expression of STAT1, restores ZO-1 junctional expression and reverses permeability. These results highlight the importance of the JAK-STAT pathway in BBB disruption in cerebral malaria and

link BBB inflammation with vascular permeability. These results are exciting given the promising results that Ruxolitinib has had in control human malaria infection studies at decreasing inflammation, vascular dysfunction while still preserving the establishment of a memory immune response. Taking this into account, and given the importance of the JAK-STAT pathway in our model, we have decided to change the title of the manuscript to “*Plasmodium falciparum* egress disrupts endothelial junctions and activates JAK-STAT signaling in a microvascular 3D blood-brain barrier model”

We have included new paragraphs describing Fig. 6 in the Results section (Lines 319-336) and in the discussion (427-441).

In extended Fig 3: the downregulation of Notch signaling pathway downstream effectors such as DLL4, JAG1/2 shown in the heatmap is not very convincing, same for ANGPT1/2

Thank you for pointing out these genes. We have removed ANGPT2 from the graph in Extended Data Fig. 3 and from the text in the manuscript as it only shows minor downregulation and is not a focus of our paper.

All of the other changes shown in the heatmap of extended Fig 3 are significant. We provide below the gene expression violin plots including FDR and LogFC values for DLL4, JAG1, JAG2, ANGPT2 (in endothelial cells) and ANGPT1 (in pericytes) as a clarification.

In Fig4: when the authors use a different life stage of the parasite (e.g., trophozoite) they observe downregulation of genes regulating blood vessel development such as APLN, ROBO4, etc; However, in the egress media model, they observed different pathways namely NOTCH, JAK/STAT, VEGF, is there any regulation of downstream effectors of these pathways with the trophozoite/schizonts model?

Thank you for pointing out these differences. We observe a significant downregulation of genes regulating blood vessel development (APLN, ROBO4) in the egress and the schizont condition, however not in the trophozoite condition. Our interpretation is that iRBC rupture is necessary for dysregulation of transcripts associated with vascular development, but this pathway is not activated by cytoadhesion. The downregulation of NOTCH pathway downstream effectors (DLL4, JAG1/2) could only be observed in egress media and not in schizonts. These results suggest that dysregulation of these pathways requires a high concentration of egress products which did not seem to be sufficient in the schizont condition with localized egress. The same might be true for the other two pathways mentioned. We include below a PROGENy pathway analysis of the trophozoites and schizont conditions showing no strong changes in JAK/STAT or VEGF signaling. We agree with the reviewer that future studies are necessary to clarify how these specific signaling pathways are mediated by the malaria parasite and its egress products, but we believe that this is out of the scope of this work.

CD8+ T cells have been identified as key players in the disruption of the endothelial barrier and targeting these cells specifically is currently in clinical trials for cerebral malaria in African children. How do you integrate or reconcile the current findings with this well-established mechanism? Do you envision them as independent events, or is the uptake and presentation of parasite products via MHC Class I perhaps the target of TCR-mediated destruction of endothelial cells by CD8+ T cells? This concept needs to be discussed.

Thank you for the suggestion. We agree with the reviewer’s hypothesis and envision it as a series of sequential events, with MHC class I induced by parasite egress products followed by the recruitment CD8+ to vascular regions that are already damaged. The activation of MHC class I in astrocytes and pericytes suggest that this recruitment could be perivascular. We have now included this plausible disease mechanism in the discussion in the paragraph that covered internalization of malaria products and antigen presentation.

Lines 416-421:

“Our findings and others suggest a potential mechanism for leukocyte recruitment not only intravascularly by endothelial cells, but also at the brain perivascular space, aligning with recent observations of vascular and perivascular accumulation of CD8+ T-cells in the brain

microvasculature in post-mortem samples of CM patients. In this scenario, our results suggest that blood-brain barrier cells might be able to present internalized parasite material to CD8+ T-cells, which could potentially amplify damage at the vascular and perivascular space.”

Minor comments:

In abstract provide abbreviation form of blood brain barrier (BBB) when first used in line 24

Line 23

“We developed a 3D blood-brain barrier (BBB) model [...]”

Describe in full EPCR and ICAM1 (line 47, Main)

Line 50-51

“binding to endothelial receptors, endothelial protein C receptor (EPCR) and intercellular adhesion molecule 1 (ICAM-1), [...]”

Fig 2K: add in result section that permeability was assessed by FITC-Dextran 70KDa (although it is written in the legends, it will be easier for the reader to know what assay was used) Line 147

Thank you for bringing this to our attention. We have edited the text as follows:

Line 154

“Baseline microvascular permeability was measured by 70 kDa FITC-dextran perfusion [...]”

In Figure 3e, please provide the Mann-Whitney p values.

We have added the p values to Figure 3e, as suggested by the reviewer.

In Figure 3j, the legend indicates magenta for ICAM-1, but there is no magenta visible. Instead, there is a light blue stain that appears to mark nuclei – is this DAPI? Also, the prominent feature appears to be loss of nuclei. Also, the magnification appeared to change. Please examine this figure closely and correct any issues.

Thank you for spotting that the DAPI label was missing in Fig. 3j. We have now moved this figure to Extended Data Fig. 4d to get space for the new IF staining of astrocytes that the reviewer suggested in the major comments. On the astrocyte monolayers experiments, cells were labelled with ICAM-1 (in magenta), which is barely expressed. In this image, we wanted to show that incubation with egress media did not induce a clear astrocytic activation, as opposed to incubation with a cytokine cocktail, as it was previously shown in Extended data Fig. 4d. We hope that getting the egress-media astrocyte stimulation and the cytokine controls in the same figure increases the clarity.

The apparent loss of nuclei in the iRBC-egress media condition is likely due to cell death, as both conditions were initially seeded with the same number of cells. This observation aligns with our scRNA-seq data, which show increased p53 expression specifically in astrocytes, consistent with cellular stress or apoptosis. Unfortunately, we could not expand our interest on astrocyte apoptosis in the 3D model due to challenges with the immunofluorescence. Regarding the concern raised by the reviewer on the image magnification, we have double checked and confirmed that all images were acquired using the same settings and magnification.

While the egress media reflects a calculated parasite density, what do the authors speculate will be the concentration of egress products in the blood circulation, and might there be local variation in this concentration? In other words, how closely do these experimental conditions reflect the levels that endothelial cells are exposed to in vivo during a malaria infection?

Thank you for bringing up this point. As stated in the manuscript, for egress media preparation, the concentration was adjusted to approximately 50×10^6 ruptured iRBC/mL, which corresponds to circulating parasitemia values frequently found in malaria patients (5×10^4 parasites/ μ L). While we do not expect all parasites to rupture simultaneously in the bloodstream, we believe that in microvascular regions of iRBC vessel obstruction, highly prevalent in autopsy studies, this concentration quite likely reflects the levels of egress products to which endothelial cells are exposed.

In some experiments PECAM1 is used for normalization. It is important that PECAM1 expression itself is not affected by the egress media.

Thank you for raising your concern. We have only used PECAM1 for normalization in the qPCR experiments in Extended Data Fig. 1b to characterize the model. This experiment was only used to compare gene expression levels in our 3D-BBB over different days in culture in control conditions. iRBC-egress media was not used in this experiment.

Reviewer #3 (Remarks to the Author):

The manuscript presents a 3D blood-brain barrier model to understand how *P. falciparum* parasites affect brain vasculature using primary human cells (HBMECs, astrocytes, and pericytes) with basic validation through immunofluorescent staining of cell-specific markers and tracking via lentiviral labeling. The key findings revealed that when exposed to *P. falciparum* egress products, the BBB model showed downregulation of critical endothelial junction and vascular development genes, leading to increased microvasculature permeability. The study also found that while short-term exposure to infected red blood cells caused more localized changes, both conditions compromised BBB integrity. This timely and necessary study offers valuable insights into the mechanisms of BBB disruption in cerebral malaria. Despite the availability of effective drugs, the lack of adjunctive therapy to prevent severe complications of cerebral malaria remains a critical limitation in recovery for survivors of this deadly disease.

The innovative bioengineered 3D model demonstrates functional barrier properties and cell-type-specific responses. However, the main concern is that the authors do not mention any limitations of the primary cells used in the study. Overall, the manuscript is well-written and logically organized. The authors have done a good job characterizing iRBCs and addressing barrier permeability questions, and the visualization and quantification of cell-specific markers are comprehensive.

We thank the reviewer for the constructive feedback and for highlighting the strengths of our model and experimental approach. We appreciate the comments on the relevance of the study and the characterization of cell-specific responses and barrier function.

We have addressed the reviewer's concerns about the limitations of the primary cells in the following paragraph.

Specific points to consider:

As with any study utilizing HBMECs, additional information about the primary cells used would be helpful. Were these pooled from multiple donors? Was the same batch of cells used for all experiments? This is unlikely, considering the amount of work presented. In that case, was some validation done between lots of cells used? These additional details would provide crucial context for the reproducibility and reliability of the findings.

Thank you for bringing up this topic. We are happy to provide clarification on the primary cells used for this study. In all the experiments performed in this manuscript, we used two batches of endothelial cells (HBMEC) (Lot 376.05.04.01.2F or 376.11.04.01.2F). All experiments with astrocytes (HA) and pericytes (HBVP) were conducted using a single batch of each.

All scRNAseq experiments were performed using the same batch of HBMEC (376.05.04.01.2F), while a different batch was used for permeability assays to test barrier properties (376.11.04.01.2F). IF staining was performed on microvessels including either of the two HBMEC batches. We have provided further details about the cells lots in the Methods section:

Lines 465-466

“Primary human brain microvascular endothelial cells (HBMEC, Lot 376.05.04.01.2F or 376.11.04.01.2F, Cell Systems) were cultured in endothelial cell growth medium-microvascular (EGM-2MV, Lonza) up to passage 8.”

Furthermore, we have performed permeability experiments to confirm that the two batches of HBMEC presented comparable barrier properties in our 3D-BBB model, which we had previously only tested on 2D monolayers. This experiment shows that 3D-BBB devices both cell batches present significant lower permeability ratio compared to those treated with iRBC egress media. The results of these experiments are now included in Extended Data Fig.2c, along with another uRBC control suggested by Reviewer 2, and were added to the text:

Lines 159-161

“Notably, no significant differences in microvascular permeability ratio were observed between the two HBMEC donors used in this study (Extended data Fig. 2c), nor in 3D-BBB microvessels incubated with uRBC media control (Extended Data Fig. 2d).”

The study significantly advances BBB modeling for cerebral malaria with 3D microvascular fabrication. The authors do address the potential benefits of iPSC-derived cells for better standardization in their discussion. The use of such a model is largely absent in cerebral malaria. They also mention the report by Lu et al. (2021), which challenges the iPSC-BBB model proposed by Lippmann et al. (Nat. Biotech, 2012). However, the iPSC model has since been further refined, evaluated, and widely applied over the last decade. If the authors want to mention the Lu et al. study as a limitation of iPSCs, they should note that the model proposed by the group utilized the overexpression of ETS transcription factor transgenes from lentiviral vectors and lacked several critical barrier phenotypes. iPSCs have other limitations that the authors can address.

We appreciate the suggestion of the author to include further limitations of iPSC-derived cells. The reason we did not go into detail into this topic is that this is the major focus of another project of our research group. Nevertheless, we have extended on the topic in the discussion, as follows:

Line 445-453

While the barrier properties of our 3D-BBB microvessels could be enhanced using induced pluripotent stem cell (iPSC)-derived brain endothelial cells⁶⁰, concerns have been raised about their epithelial-like characteristics⁶¹, particularly in early protocols. Recent advances have shown that overexpression of ETS factors, such as ETV2, FLI1, ERG⁶¹, or FOXF2 and ZIC3⁶² can shift these cells towards an improved vascular and brain vascular phenotype, respectively. As other differentiation mechanisms are currently being developed for the generation of iPSC-derived BBB models^{63,64}, iPSCs still remain a promising tool for generating in vitro models of the microvasculature of the brain and other organs affected during *P. falciparum* infection⁶⁵.

Minor points:

Additional details on optimizing cell ratios for constructing the 3D model, including more information on the QC metrics for the 3D model, would be beneficial. This information, combined with previously mentioned data on the reproducibility of primary cell donor findings, would address any validity concerns about the study.

When optimizing the 3D-BBB model, we have tested different astrocyte-to-pericyte ratios to choose a cell combination that would enhance barrier properties. To address this point, we have included the qPCR results comparing the different cell ratios in Extended Data Fig.1b.

Furthermore, we have edited the manuscript, as follows:

Line 88-91

“Quantitative RT-PCR measurements comparing two different astrocyte-to-pericyte ratios revealed that co-culture with astrocytes and pericytes at a 7:3 ratio increased the expression of endothelial BBB-specific markers over time, [..].”

Line 499-501

“A 1:1 astrocyte-to-pericyte ratio (2.5x10⁵ HA/mL(collagen): 2.5x10⁵ HBVP/mL(collagen)) was initially used to compare the expression level of BBB-specific markers by quantitative polymerase chain reaction.”

Reviewer #4 (Remarks to the Author):

Thank you for your suggestions. We have addressed this reviewer's comments in previous sections.

REVIEWERS' COMMENTS

Reviewer #1 (Remarks to the Author):

The authors have addressed all of my comments. I have no further requests or comments.

We thank the reviewer and are glad that the revisions have addressed their concerns.

Reviewer #2 (Remarks to the Author):

We thank the authors for the careful consideration of all the reviewers' suggestions and questions. Several of the key points we raised were directly implemented in the revision. Notably, the authors incorporated the suggestion to test the effect of the JAK/STAT inhibitor (ruxolitinib) in their 3D system with egress media. Their new Figure 6 provides compelling data showing that ruxolitinib helps restore tight junction integrity and improves barrier permeability—an important advancement in understanding how this pathway could be targeted therapeutically in cerebral malaria (CM). We endorse the authors' revised the title that better reflect the central role of JAK/STAT signaling.

Another critical point raised concerned the validation of their ultrastructural findings. We recognize the new LAMP1 staining experiments to confirm lysosomal uptake of parasitic material by endothelial cells, strengthening interpretation of the TEM data. We appreciate the additional findings on astrocyte behavior in the presence of egress media, which suggests a mechanical stabilizing role at sites of endothelial disruption.

Overall, the authors' additional efforts have significantly strengthened this work and its translational potential. The revised paper now more clearly highlights the therapeutic relevance of JAK/STAT inhibition in CM, particularly the potential to repurpose ruxolitinib in combination with antimalarials to restore blood-brain barrier (BBB) function. While further in vivo studies are still necessary to assess neurobehavioral outcomes, the 3D model remains a robust and promising tool for therapeutic testing.

Nafiisha Genet, PhD (UVA)
Hans Ackerman, MD DPhil (NIAID)

We sincerely thank the reviewers for their thoughtful feedback and are pleased that they found the additional experiments of interest and that these revisions have enhanced the impact of the manuscript.

Reviewer #3 (Remarks to the Author):

The authors have adequately addressed my concerns about the previous iteration of the manuscript. I appreciate the authors' detailed response to each review question and the inclusion of useful additional data.

We appreciate the reviewer's careful assessment and are glad that the additional data and responses were well received.

A couple of important new questions arise based on new data included in the revised manuscript.

A point-by-point response to the new questions is provided below.

1. The earlier iteration discussed differential mRNA expression only. In the revised manuscript, the newly added Figures 6a-c raise questions about TJ protein expression and localization by IF.

The authors were able to stain for ZO-1, but not for Claudin-5 or occludin, and offered a few explanations in the rebuttal, but have not addressed a critical missing data point at all in the manuscript beyond mentioning increased mRNA expression.

A known limitation of primary HBMECs is the reduced expression of claudin-5 compared to microvessels, which is a key driver of the low junctional tightness of these primary cells. Gericke et al. (2020) is more recent among the studies attempting to address the limitation of low Claudin-5 expression in HBMECs.

Figure 6 b shows that Ruxolitinib increases STAT1 expression and rescues ZO-1, but without Claudin-5, the specific ability to restore BBB integrity is questionable. ZO-1 is a scaffolding protein, Claudin-5 is the primary transmembrane protein of the BBB, and occludin modulates barrier stability.

While the focused title highlighting the activated JAK-STAT pathway is reasonable, the study has not shown sufficient evidence of BBB junctional disruption. The claim that Ruxolitinib preserves BBB integrity would be strengthened by post-treatment assessment of claudin-5 localization and expression.

Thank you for raising this concern. As suggested by the editors in the Author Checklist, we have tempered our conclusions about Ruxolitinib effect and mentioned the limitations of our primary cells in terms of Claudin5 and Occludin staining in the manuscript.

Lines 33-34:

"Treatment with the JAK-STAT inhibitor Ruxolitinib prevents the increase in permeability induced by *P. falciparum* egress products.

Line 447-450:

"Although presenting improved barrier properties, our model is still far away from recreating BBB physiological permeability rates, which is likely related to the inherently low expression of tight junction proteins like claudin-5 and occludin in primary endothelial cells^{60,61}".

2. Relatedly, the absence of EPCR/ICAM-1 binding validation is a major omission. The

earlier draft included ICAM-1 expression, which I misunderstood to be on the surface of HBMECs. A response to a reviewer's query clarified that it was about astrocytes.

We appreciate the reviewer highlighting the importance of validating EPCR/ICAM-1 binding on the *P. falciparum* lines used in this study. Perfusion with *P. falciparum* clonal lines expressing a single PfEMP1 was done at low parasite cycle, short after. The qPCR below shows the expression of HB3var03, with predicted dual binding to both ICAM-1 and EPCR. This analysis is routinely performed in our laboratory, as part of our experimental workflow. We have now included the corresponding graph in the Supplementary Fig. 5a and formally referenced it in the results and methods sections (Line 536-546).

In addition, the dual binding phenotype of HB3var03 on 3D microvessels has been shown in the past by us (Bernabeu et al., 2019), where reduced sequestration was observed following treatment with ICAM-1 or EPCR-blocking antibodies, confirming its ability to bind both receptors. Furthermore, we have recently shown the reproducibility of the binding phenotype and confirmed it since the group move to EMBL Barcelona through inhibition with anti-PfEMP1 monoclonal antibodies targeting the domain CIDR α 1.4, responsible for binding to EPCR (Reyes et al., 2024)

3. In response to Reviewer 2's comment about egress media as a correlate for parasite density. The authors suggest 5×10^4 ruptured iRBCs/uL correspond to about equal numbers of circulating parasitemia values of 5×10^4 parasites/uL or 1%, which are frequently found in patients. This is mathematically consistent if we assume 1 iRBC = 1 parasite. However, late-stage schizonts contain anywhere from 16 to 24 merozoites.

The authors' referenced manuscript (Long et al., 2024, bioRxiv preprint) details the methodology for blocking iRBC egress and the use of late-stage schizonts. There is no mention of testing higher or lower dilutions of iRBCs used, so the concentration used is only justified by the math above, which does not account for merozoites.

Even accounting for circulating parasitemia in clinical infections being an underestimate of the actual parasite density (due to sequestered parasites), using mature schizonts, the egress media likely delivered a significantly higher parasitemia than would be seen in the most severe clinical cases.

We thank the reviewer for raising this concern. As suggested by the editors, we have now clarified that the parasite density in iRBC-egress media is approximate and may exceed levels observed in clinical infection.

Line 112-113:

“We estimate that this concentration of products is equivalent to 5×10^4 parasites/ μL , levels often found in CM patients⁵; however, this value is an approximation and may exceed peripheral parasite densities encountered in clinical infections.”

4. With the additional Ruxolitinib experiments, the authors can show some other measures of baseline vs. iRBC exposed vs. drug

We appreciate the reviewer’s suggestion regarding additional measures in the Ruxolitinib experiments. The current data serve as a proof of concept, demonstrating that the inhibition of the JAK/STAT pathway contributes to endothelial barrier stabilization upon exposure to iRBC egress media. While further experiments are indeed valuable, they are beyond the scope of this manuscript and will be pursued in future studies. As noted in our previous in comment 2, we have also moderated the conclusions from the Ruxolitinib experiments in accordance with the editors’ recommendations.

Reviewer #4 (Remarks to the Author):
